



# Classification of synoptic circulation patterns with a two-stage clustering algorithm using the structural similarity index metric (SSIM)

Kristina Winderlich, Clementine Dalelane, Andreas Walter

Climate and Environment Consultancy Department, Deutscher Wetterdienst (German Meteorological Service), Offenbach am Main, 63067, Germany

*Correspondence to*: Kristina Winderlich (Kristina.winderlich@dwd.de)

**Abstract.** We develop a new classification method for synoptic circulation patterns with the aim to extend the evaluation
routine for climate simulations. This classification is applicable for any region of the globe of any size given the reference data. Its unique novelty is the use of the structural similarity index metric (SSIM) instead of traditional distance metrics for cluster building. This classification method combines two classical clustering algorithms used iteratively, hierarchical agglomerative clustering (HAC) and k-medoids, with the only one pre-set parameter - the threshold on the similarity between two synoptic patterns expressed as the structural similarity index measure SSIM. This threshold is set by the user to
imitate the human perception of the similarity between two images (similar structure, luminance and contrast) and the number of final classes is defined automatically.

We apply the SSIM-based classification method on the geopotential height at the pressure-level of 500hPa from the reanalysis data ERA-Interim 1979-2018 and demonstrate that the built classes are 1) consistent to the changes in the input parameter, 2) well separated, 3) spatially and temporally stable, and 4) physically meaningful.

We use the synoptic circulation classes obtained with the new classification method for evaluating CMIP6 historical climate simulations and an alternative reanalysis (for comparison purposes). The output fields of CMIP6 models (and of the alternative reanalysis) are assigned to the classes and the quality index is computed. We rank the CMIP6 simulations according to this quality index.



## 1 Introduction

Research institutions around the world conduct climate studies and share their knowledge with the society and policy makers
through The Intergovernmental Panel on Climate Change (IPCC, www.ipcc.ch). The climate simulations used in the IPCC
reports are available to other scientists, besides those who run the models, through the Coupled Model Intercomparison
Project (CMIP, www.wcrp-climate.org/wgcm-cmip). The first two phases (CMIP1 and CMIP2) of this initiative addressed
the ability of numerical climate models to simulate the present climate and to respond to an increase of carbon dioxide
concentration in the atmosphere (Meehl et al., 1997; Meehl et al., 2000). The extended follow-up phase CMIP3 (Meehl et al.,
2007) provided output of coupled ocean-atmosphere model simulations of 20th-22nd century climate for the 4th Assessment
Report (AR4) of IPCC (www.ipcc.ch/report/ar4/syr/). As the number of climate simulations in subsequent projects CMIP5
(Taylor et al., 2012) and CMIP6 (Eyring et al., 2016) continued to increase, new requirements on the "quality" and
"reliability" of such simulations emerge.

Hannachi et al (2017) emphasized the importance of the correct representation of weather regimes, their spatial patterns, and
persistence properties in global circulation models as they could properly simulate the climate variability and long-term
climatic changes under an external forcing such as, for example, the global warming. However, traditional techniques for
model evaluation mainly focus on individual variables and/or derived indices and do not take into account, how well models
simulate synoptic weather patterns and their frequencies of occurrence (Díaz-Esteban et al., 2020). As some studies have
already demonstrated that the performance of a model varies as a function of weather regimes (Díaz-Esteban et al., 2020;
Nigro et al., 2011; Perez et al., 2014; Radić and Clarke, 2011) we can no longer ignore the model dynamics in the evaluation
routine. Therefore, we propose to examine the correctness of the representation of synoptic patterns in climate simulations
additionally to commonly evaluated variables such as temperature and precipitation.

The atmospheric circulation is a continuum that gradually changes and its dynamics can be described by a finite number of
representative "states"/"typical patterns" i.e. classes. Hochman et al. (2021) proved that such representation of the
atmosphere by quasi-stationary circulation patterns, often also termed as weather regimes, is a physically meaningful way to
describe the atmosphere (and not only a useful statistical categorization as it may be argued). Muñoz et al. (2017) also
suggested using the weather-typing approach to diagnose a range of variables in a physically consistent way helping to
understand the causes of model biases. For evaluation purposes, any climate model simulation can be represented as a
sequence of typical synoptic situations, previously classified. Common variables used for representing the synoptic
circulation are the sea level pressure, geopotential heights and wind vector fields. Statistical measures, such as frequency and
duration of each class, computed from the assigned sequence can be evaluated against reference data derived, for example,
from a reanalysis.

There is no objective number of the classes for describing the moving atmosphere. A set of classes can be determined
subjectively by an expert, as the well-known Hess-Brezowski Grosswetterlagen (Gerstengarbe and Werner, 1993; Hess and
Brezowsky, 1952; James, 2006) or the Lamb weather types (Lamb, 1972), or using an automated classification method.



Multiple different synoptic classifications have been developed over the years as summarized by Yarnal et al. (2001) and Huth et al. (2008). An overview and systematization of existing classification methods for synoptic patterns was compiled in a joint effort of multiple European Institutions in a COST Action 733 and summarized in the final project report (Tveito et al., 2016). However, we consider none of the methods from this report suitable for our purpose of synoptic pattern
classification because these methods either use a debatable (in our opinion) initialization routine or a suboptimal (in our opinion) distance metric for cluster building.

We introduce a new two-stage clustering algorithm for classification of the synoptic circulation patterns. The novelty of this method consists of the following features: it uses a similarity metric instead of a distance-metric, it represents clusters by their medoids instead of centroids, and it uses an iterative combination of the hierarchical agglomerative algorithm with a
partitioning k-medoids algorithm to determine the number of clusters automatically. This clustering algorithm does not need an initial distribution of elements and gradually continues building and reviewing clusters until there is no more clusters to be built and reviewed according to a given threshold of similarity.

We demonstrate that the new classification produces a set of well-separated classes, not necessarily of similar size, consistent (small changes in the pre-set parameters do not alter classes strongly), stable in various spatial resolutions and
data volumes, and physically interpretable.

The paper is structured in the following way: 1) data and domain description, 2) description of the clustering method, 3) presentation of resulting classes and 4) use of the derived classes for evaluating CMIP6 climate simulations computing the quality index, 5) our conclusions and an outlook for future applications.

**2 Data**

We use the Reanalysis ERA-Interim (Dee et al., 2011) for period of 1979-2018 as a realistic historical representation of the atmospheric circulation in Europe. Simulated synoptic regimes are represented by the geopotential height (*zg*) at the pressure level of 500 hPa sampled daily at 12:00 UTC and spatially on a grid of 2°x3°. The coarse-scale sampling is sufficient due to the fact that the synoptic-scale 500-hPa geopotential height does not require high resolution to reproduce the key physical mechanisms associated with (Muñoz et al., 2017). The chosen domain (Fig. 1) has 22x22 grid points with the lower left
corner at (20°W, 29°N).



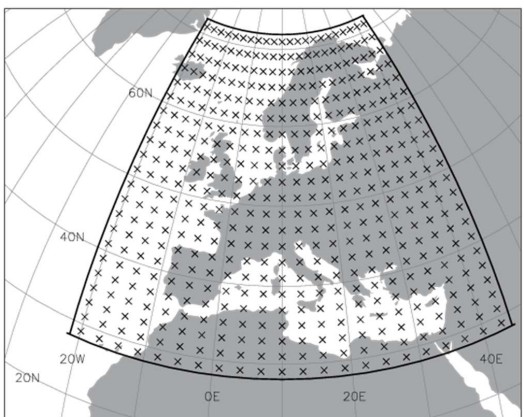

**Figure 1: Domain for classification of synoptic circulation patterns: crosses show sample points every 2° in latitude and every 3° in longitude directions, 22x22 grid points in total. The solid black line shows the outer edge of the domain.**

Some typical synoptic patterns may occur in different seasons but should be grouped into one class. To allow this, we pre-process the original geopotential height fields ($zg$): remove the seasonal amplitude from the original daily data and normalize the resulting fields by the daily standard deviation as in Eq. (1):

$$zg_a = (zg - zg_{mean})/zg_\sigma \qquad (1)$$

The mean $zg_{mean}$ and the standard deviation $zg_\sigma$ are calculated for each grid point and for each day of the year from the 40-years of ERA-Interim data; both fields are smoothed in time with 151-days running average.

We use the resulting from the classification set of synoptic patterns to evaluate 32 global circulation models (Table 3) available from the Coupled Model Intercomparison Project Phase 6 (CMIP6, https://www.wcrp-climate.org/wgcm-cmip/wgcm-cmip6, (Eyring et al., 2016)) and compute quality indices for each model simulation. The models were chosen for the historical period 1950-2014, preferably simulation version r1i1p1f1 when available or r1i1p1f2/r1i1p1f3 otherwise. Additionally, we evaluate the alternative reanalysis data NCEP1 (Kalnay et al., 1996) for a comparison to the evaluation of the CMIP6 models. Assuming that the alternative reanalysis captures the synoptic circulation better than any unconstrained global circulation model, this evaluation gives the estimate for the best quality index possible.

## 3 Method

A very frequently used approach for identifying circulation regimes is to apply the k-means clustering algorithm to the synoptic circulation data (an overview can be found in the project report COST Action 733 by Tveito et al. (2016)). The k-means method partitions the input data into k clusters, such that each data element belongs to the cluster with the nearest



centroid minimizing within-cluster variances; the k-means method is simple and always converges to a solution. Although k-means and its multiple variants are commonly applied in the field of the atmospheric science, they exhibit serious limitations for our aims: 1) they use centroids (means) to represent classes, 2) they require a pre-specified number of classes and 3) they

use structure-insensitive distance metrics (e.g. mean square error MSE) for the optimization of the element assignment among classes. The k-means clustering assigns every data element to the cluster center that is closest to it, if only by a small margin. This makes the method sensitive to noise in the data and may lead to an assignment of a data element to a structurally dissimilar cluster center (Falkena et al., 2021).

Following the previous considerations, we made three essential decisions to modify the classic k-means algorithm in order to

construct an algorithm better suitable (from our perspective) for building classes of synoptic patterns.

Decision 1: use medoids to represent classes. Using the centroids to represent classes leads to two serious problems: the mean of the fields, each of which describes a meaningful synoptic circulation, is not always interpretable i.e. may represent an unrealistic synoptic situation; the mean field may serve as an "attractor" fields that are dissimilar to each other ("snowballing"). Therefore, we propose to use medoids for class representation. A medoid is the element of the class with

the smallest dissimilarity to all other elements in this class. Each medoid itself is part of the data and represents a physically realistic synoptic situation. Additionally, using medoids makes the classification algorithm (k-medoids) less sensitive to outliers helping to avoid "snowballing" effects.

Decision 2: use a two-stage algorithm. There are multiple ways of defining the number of classes for a k-medoids algorithm (similarly to k-means) ranging from a random guess to the analysis of the data based on principal component analysis PCA,

also known as empirical orthogonal functions, Huth (2000). Lee and Sheridan (2012) suggested the initialization of the clustering algorithm by selected PCAs. The reason for this statement was the common (naïve) assumption that the first few modes returned by PCA were physically interpretable and should match the underlying signal in the data. However, Fulton and Hegerl (2021) tested this signal-extraction method and demonstrated that it has serious deficiencies when extracting multiple additive synthetic modes: false dipoles instead of monopoles, which may lead to serious misinterpretation of

extracted modes. Fulton and Hegerl (2021) also found that PCA tends to mix independent spatial regions into single modes. Therefore, we back off using the PCA-based initialization of the clustering algorithm and employ another classic clustering algorithm, hierarchical agglomerative clustering (HAC), for initializing the k-medoids. We build an algorithm consisting of two parts – HAC and k-medoids – that are called iteratively. The HAC-algorithm combines similar clusters and, subsequently, the k-medoids algorithm reviews them. The two-stage algorithm stops when no similar clusters are left.

Decision 3: use an alternative similarity measure. The mean squared error (MSE) and the Pearson correlation coefficient (PCC) are probably the dominant quantitative performance metrics in the field of model evaluation and optimization. However, Wang and Bovik (2009) demonstrated that the MSE has serious disadvantages when applied on data with temporal and spatial dependencies and on data where the error is sensitive to the original signal. Mo et al. (2014) in turn demonstrated that the PCC as a metric is insensitive to differences in the mean and variance. However, atmospheric data (pressure,

geopotential, temperature fields) often reveal dependencies in time and space, as well as shifts in the mean and differing



variances. Both studies mentioned above (Mo et al., 2014; Wang and Bovik, 2009) recommend using an alternative measure to signal/image similarity, the Structural Similarity (SSIM) index, to quantify the goodness of match of two patterns. The SSIM (Wang et al., 2004) simulates the human visual system that "recognizes" structural patterns and error-signal dependencies, and shows a superior performance as a similarity measure over the MSE and PCC.

Considering the previous arguments, we develop a new classification method for synoptic circulation patterns that uses the similarity measure SSIM. This method is a new two-stage classification algorithm that builds classes of synoptic circulation patterns according to a given threshold on the similarity without pre-defining the number of classes. The method uses medoids for representing classes, instead of the commonly used centroids, and thereby makes the classification algorithm less sensitive to outliers (anomalous/untypical synoptic patterns).

## 3.1 Similarity measure for synoptic patterns


We use the Structural Similarity index SSIM (Wang et al., 2004) for measuring the similarity between synoptic pattern (SP) fields represented by the geopotential height $zg_a$–anomalies. These fields are highly structured images, meaning that the sample points of these images have strong neighbour dependencies, and these dependencies carry important information about the structures of the highs and lows in the field. The SSIM incorporates three perception-based components of image

difference: structure (covariance), luminance (mean) and contrast (variance):

$$SSIM(x,y) = \frac{(2\mu_x\mu_y + c_1)(2\sigma_{xy} + c_2)}{(\mu_x^2 + \mu_y^2 + c_1)(\sigma_x^2 + \sigma_y^2 + c_2)} \tag{2}$$

where

$x, y$ - images,

$\mu_x, \mu_y$ - average values for $x$ and $y$,

$\sigma_x, \sigma_y$ - variance for $x$ and $y$,

$\sigma_{xy}$ - covariance of $x$ and $y$,

$c_1 = (k_1 L)^2$, $c_2 = (k_2 L)^2$ - stabilizing constants for weak denominator,

$L = 20$ – dynamic range of $zg_a$-values,

$k_1 = 0.01$, $k_2 = 0.03$.

Each SP is treated as a two-dimensional image. For each pair of images the SSIM-value is computed. The SSIM takes value 1 only for two identical images; a value less than 1 identifies some difference between two images. Typically, the SSIM-value is computed for multiple sliding windows inside the image. But for simplicity here, only one SSIM-value is computed

for the whole domain (Fig. 1). As the selected domain is relatively large and extends to high latitudes, areal weighting was applied to all fields before computing the SSIM.



### 3.2 Classification method: two-stage clustering algorithm

The two-stage clustering algorithm combines two clustering methods - the hierarchical agglomerative clustering (HAC) and the k-medoids clustering - in such way that the output from the first is used as input into the second (Fig. 2). Medoids are used for cluster centers. A medoid is the element of the cluster with the smallest dissimilarity to all other elements in the cluster. The HAC is a very flexible clustering method that can use any distance or [dis]similarity measure as it allows different rules for aggregating data into clusters(Schubert and Rousseeuw, 2021). The k-medoids algorithm is then initialized with the medoids built by HAC and rearranges data elements between these clusters (an operation that HAC cannot do) in

order to maximize the within-cluster homogeneity. It builds clusters (similarly to the wide-known method of k-means) using the medoid-prototypes and an arbitrary [dis]similarity measure for cluster similarity (D'Urso and Massari, 2019; Schubert and Rousseeuw, 2021).

The two-stage clustering inherits the strengths of both contributing algorithms. Initially each data element represents its own cluster. The first step, HAC, determines the number of clusters and their medoids without a prior estimation. For each two

clusters merged into one, the medoid of the new cluster is recomputed. The threshold on the SSIM-value for merging similar clusters $TH_{merge}$ is set by the user. The second step, k-medoids, in few iterations produces optimized clusters using the "seed" of cluster medoids delivered by the first step.

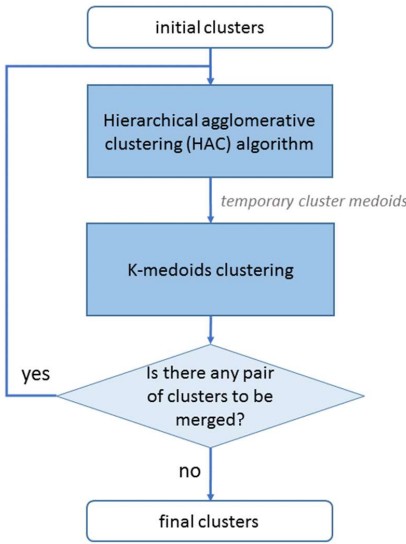

**Figure 2: Flow diagram of the two-stage clustering algorithm.**



At each iteration of the two-stage clustering, two steps are done as illustrated in Fig. 2 in the following way:

1. **The 1st Step: HAC (merging clusters):**

1.1.     Clusters with sufficient similarity $SSIM>TH_{merge}$ are merged to create bigger clusters: clusters with higher similarity are merged prior to those with lower similarity.

1.2.     Temporary cluster medoids are recomputed.

2. **The 2nd Step: k-medoids (recompose clusters):**

2.1.     Temporary cluster medoids from the first step are used to initialize the k-medoids clustering algorithm.

2.2.     Each data element is assigned to the cluster with the most similar medoid.

2.3.     Cluster medoids are recomputed.

2.4.     K-medoids clustering is repeated until an optimum (for the given number of medoids!) distribution of all data elements is achieved.

Both steps are repeated until no cluster pair is sufficiently similar to be merged.

The presented classification method, as any other classification method, requires some pre-set parameters. The final number of clusters (that later build classes) produced by the two-stage clustering algorithm depends on the threshold $TH_{merge}$ for cluster merging and, eventually, on the amount of data to be clustered. Although the choice of $TH_{merge}$ is crucial, there is no statistical or analytical formula for computing this threshold. $TH_{merge}$ can only be chosen subjectively by comparing pairs of images and asking an observer about his/her perception of similarity. Examples of "similar" synoptic patterns are shown in Fig. 3. We analyzed multiple pairs of SP-images and, based on the personal perception of similarity (our own as well as of persons not evolved into the development of this classification method!), estimated the threshold value $TH_{merge}$=0.45 for recognized similarity i.e. image pairs with SSIM-value less than $TH_{merge}$ being perceived as dissimilar. Figure 3 illustrates examples of similarity between three exemplary reference SP-images and arbitrarily chosen SP-images with SSIM-values of 0.75, 0.55, 0.50, 0.45, 0.40, and 0.10 to each reference. SPs with $SSIM{\geq}0.75$ are "strongly similar" to the reference. SPs with the $SSIM<0.45$ are considered "weakly similar" to the reference. SPs with $SSIM<0.10$ are "dissimilar" to the reference.





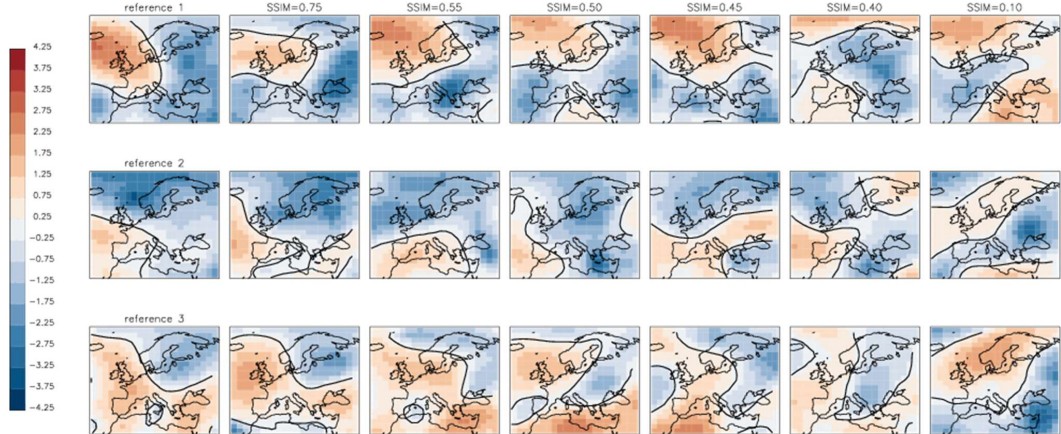


**Figure 3: Examples of three synoptic patterns $zg_a$ (left column "reference"). Each row contains examples of alternative synoptic patterns with the SSIM-value to the "reference".**

The pre-set threshold for merging clusters $TH_{merge}$ is crucial for the two-stage clustering algorithm. It is chosen subjectively and can vary. The definition of this threshold implies that a reduction of its value loosens the requirement on data similarity

for cluster building and provides a smaller number of final classes. In contrary, an increase of $TH_{merge}$ stratifies the requirement on the data similarity for cluster building and, therefore, leads to a larger number of final classes. At the same time, the higher $TH_{merge}$ also loosens the requirement of separation among classes and permits a higher similarity among them. Thus varying the value of $TH_{merge}$ may be used, in some extent, to steer the clustering algorithm to produce the number of final classes in the particularly desired magnitude.

Keeping in mind the intended application (evaluation of climate models) the question arises: how many classes do we need to describe the synoptic flow? In the present study, we use 40 years of daily synoptic patterns, 14600 data elements, which is a usual number of available reference data in climate research for the industrial time. How many classes do we need to represent synoptic situations of these 40 years? Would 10 or 100 be sufficient? The answer to this question is not trivial. The number of derived classes depends on the pre-set parameter $TH_{merge}$. Whereas, values of $TH_{merge}$ smaller than 0.45 were

mainly discarded by observers, testing higher values remains reasonable. Tests of various values for $TH_{merge}$ yielded the following results: the value of 0.45 produces typically less than 60 synoptic classes, the value of 0.55 – over 100 classes. We set $TH_{merge}$=0.45, the minimum value of the considered range, for two reasons: 1) using this value produces fewer large classes, which can be meaningfully statistically analyzed (a higher threshold value would produce many classes with few members), 2) a smaller number of classes is easier to describe verbally, more intuitive to understand and visually more

separated.



### 3.3 Criteria for the evaluation of the clustering algorithm

We analyze the performance of the new method using four criteria suggested by Huth (1996): The clusters should (i) be consistent when pre-set parameters are changed, (ii) be well separated both from each other and from the entire data set, (iii) be stable in space and time, and (iv) reproduce realistic synoptic patterns.

Cluster consistency. The consistent evolution of classes implies that small changes in the pre-set parameter $TH_{merge}$ lead only to small changes in the classes. For illustrating the sensitivity of the clustering algorithm to the choice of $TH_{merge}$ it was run for three values: the reference value 0.45, and two higher values 0.50 and 0.55.

We observe that an increase in the number of classes, caused by a change in $TH_{merge}$, is realized predominantly by splitting few classes, with others remaining almost unchanged. Such evolution is difficult to quantify. The 'consistency' of the clusters

is illustrated by similarity diagrams alike to the "arrow diagrams" in Huth (1996) for the sets of classes built with varying parameters.

Cluster separability. According to the stop-criterion of the clustering algorithm, the derived classes have $SSIM< TH_{merge}$ for each pair of classes. Although the classes are represented by the cluster medoids in the clustering algorithm, it is also reasonable to require that the resulting cluster centroids (means) be at least not "strongly similar" ($SSIM<0.75$) to each other.

We compute matrices of similarities for medoids and for centroids and analyze how well the medoid-separation algorithm provides the separation of centroids in the final set of classes.

Additionally, we calculate three metrics introduced in in the COST Action 733 report (Tveito et al., 2016) for characterizing the separability and within-type variability. These metrics are not independent from each other. The separation of clusters from randomly chosen data is addressed by the comparison of the metrics calculated on the clusters to the metrics calculated

on "random groups". The "random groups" are generated for each cluster as groups of the same size but of randomly chosen data elements (one realization).

The explained variation $EV$ of the data is determined as the ratio of the sum of squares within classes (synoptic types) $WSS$ and the total sum of squares $TSS$:

$$EV = 1 - \frac{WSS}{TSS} \tag{3}$$

The distance ratio $DRATIO$ is the ratio of the mean distance between elements assigned to the same class $DI$ and the mean distance between elements assigned to different classes $DO$. The Euclidean distance is used for computing $DI$ and $DO$:

$$DRATIO = \frac{DI}{DO} \tag{4}$$

The "faster Silhouette Index" $FSIL$ is calculated for each data element $i$ from the distances to their own class ($fa_i$) and the distance to its closest neighboring class ($fb_i$). The Euclidean distance is used for computing $fa_i$ and $fb_i$ and for determining the

closest neighboring class:

$$FSIL = \frac{1}{n} \sum_{i=1}^{n} \frac{fb_i - f_i}{max(fb_i, fa_i)} \tag{5}$$



Cluster stability. The amount of input synoptic data is crucial for building the representative set of classes. In periods of only few years of data important synoptic circulations might be simply un- or under-represented and, therefore, omitted in the final set of classes. The clustering algorithm is run on a continuously increasing data amount from one to 40 years. After a

certain critical data amount is accumulated, further increase does not lead to a discovery of new classes. This demonstrates the temporal stability of the method. The minimum critical data amount is detected when the number of resulting classes "levels out" and stabilizes.

The stability of the method in space cannot be addressed by applying the clustering algorithm straightforward to the data on lower/higher spatial resolution because the pre-set threshold for cluster merging $TH_{merge}$ is not directly transferable to other

spatial grids. The reason for this is simple: a pair of images at a high resolution that appears dissimilar to an observer may have similar low-resolution prototypes (when similarity-determining details are averaged out). However, it can be required that the method determines identical types at any spatial resolution. To test this, the clustering algorithm is run on the same data but of reduced (4°x6°) and increased (1°x1.5°) spatial resolution. The retrieved classes are compared to the reference classes (2°x3°).

Cluster reproduction and representativity. The method uses medoids as cluster centers and, therefore, the resulting classes (set of medoids) are elements of the original data and are physically interpretable/plausible synoptic patterns. However, it is necessary to demand that a cluster medoid represents all cluster elements and their whole entity as a group. For each cluster, we compare the cluster center (medoid) to the cluster mean (centroid) and calculate the similarity value. Based on the similarity values we analyze the representativity of the cluster elements by the medoids. We require the "strong similarity"

between the medoid and centroid of each cluster with minimum similarity value of 0.75.

### 3.4 Statistics for model evaluation

The model output was assigned to the 43 reference classes derived from ERA-Interim and the following statistics were computed: histogram of frequencies (*HIST*) for SP-classes (year through), histograms of frequencies for each season (*HIST*$_{DJF}$, *HIST*$_{MAM}$, *HIST*$_{JJA}$, *HIST*$_{SON}$), matrix of transitions (*TRANSIT*) between available classes (frequency for each SP to

follow another SP), and probability of persistence (*PERSIST*) of each SP for 1,2, .. 25 days. For each of these seven statistics an individual quality index (*QI*) is computed. The overall quality index is then computed as the mean of the seven individual quality indices.

### 3.5 Quality index

We use the Jensen–Shannon divergence for measuring the similarity between two probability distributions $P$ and $Q$ defined

on the same probability space $\chi$.

$$JSD(P \parallel Q) = \frac{1}{2}\sum_{x \in X} P(x) \ln \frac{P(x)}{M(x)} + \frac{1}{2}\sum_{x \in X} Q(x) \ln \frac{Q(x)}{M(x)} \tag{6}$$





The probability distributions in our case are the normalized (to the sum of 1.0) frequency histograms, transitions- and persistence-matrices of the reference ($Q$) and a model ($P$). $M$ is the mean probability distribution:

$$M = \frac{P+Q}{2} \tag{7}$$

The Jensen–Shannon divergence is based on the Kullback–Leibler divergence, but it is symmetric and it always has a finite value. It is common to compute the square root of JSD as a true metric for distance:

$$J(P \parallel Q) = \sqrt{JSD(P \parallel Q)} \tag{8}$$

The quality index QI is calculated as suggested by Sanderson et al. (2015) but using the Jensen-Shannon distance as follows:

$$QI(P \parallel Q) = exp^{-a\sqrt{J(P\parallel Q)}} \tag{9}$$

Where $a$ - normalizing constant: $a$=10 for histograms and $a$=100 for matrices.

## 4 Method

### 4.1. Derived classes

The two-stage clustering applied to the 40 years of ERA-Interim data produced 43 SP-classes (Fig. 4). According to the stop-criterion of the clustering algorithm the final classes have mutual SSIM-value less than $TH_{merge}$=0.45 to each other.

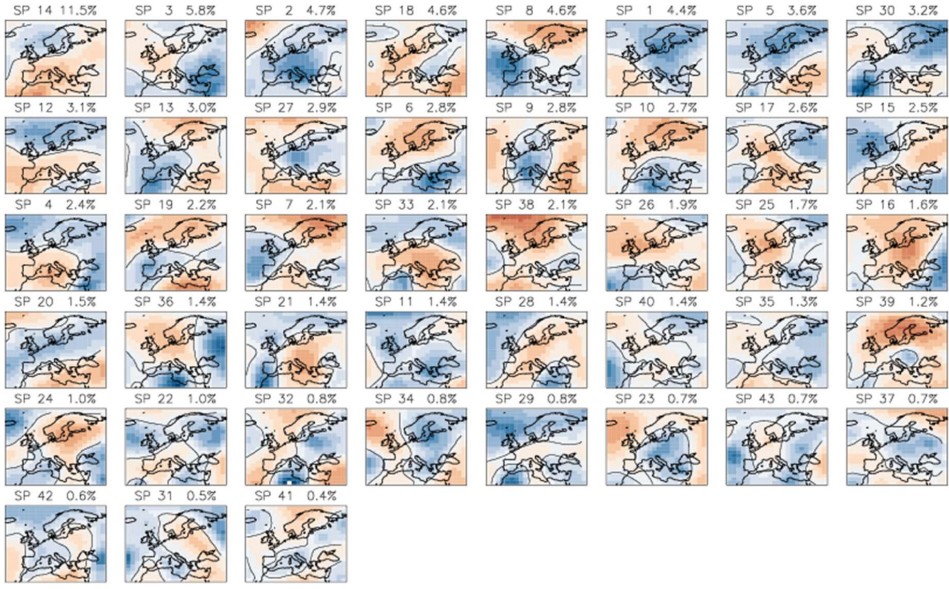

**Figure 4: Classes derived from ERA-interim reanalysis 1979-2018 with $TH_{merge}$=0.45 ordered by the frequency of occurrence (shown at the top of each plot). The legend for colour shading is the same as in Fig. 3.**





The most frequent synoptic pattern SP 14 (Fig. 5) retrieved by the new classification roughly corresponds to the westerly
flow described by Gerstengarbe and Werner (1993) as Cyclonic Westerly flow (orig. definition in German "Westlage,
zyklonal", WZ). The second most frequent pattern SP 3 resembles two flow patterns of the same study - HM (Germ.: "Hoch
Mitteleuropa") and HNA (Germ.: "Hoch Nordmeer-Island, antizyklonal"); the third most frequent pattern SP 2 represents
the flow pattern TWE (Germ.: "Trog Westeuropa"). This correspondence gives us an evidence that, albeit not tuned to and
not required to mimic semi-manual classifications, the new classification method determines not just arbitrary synoptic
patterns but those described by experts in semi-manual classifications.

The next three most frequent synoptic classes (SP 18, SP 8 and SP 1) in Fig. 5 roughly correspond to the Grosswetterlagen
NEA (Germ.: Nordostlage, antizyclonal), SEA/SEZ (Germ.; Südostlage, antizyclonal/zyclonal) and WS (Germ.: Südliche
Westlage) of Gerstengarbe and Werner (1993) and James (2015).

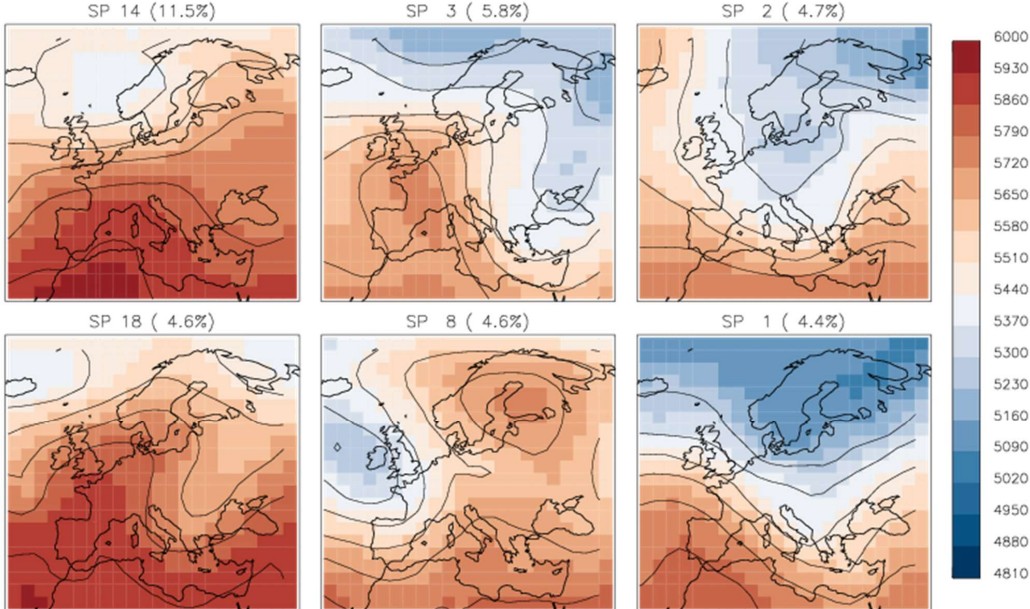

**Figure 5: Six most frequent synoptic patterns (absolute geopotential height at 500 hPa [m]: anomaly * seasonal variance + seasonal mean).**

The set of classes is further analysed on its consistency, separability, stability, and representativity of the data.





#### 4.2 Cluster consistency

The evolution of classes built with $TH_{merge}$ of 0.45, 0.50 and 0.55 is presented in the form of a diagram (Fig. 6). The classes
are derived by running the clustering algorithm on the full reference data set of 40 years. Identical classes (SSIM=1 for the
medoids) between each two sets of classes are connected with a solid black line, "strongly similar" ($0.75 \leq SSIM < 1$) classes
are connected with solid a blue line, classes with $0.45 \leq SSIM < 0.75$ are connected with a thin grey line. The solid lines in Fig.
6 shows multiple classes that are simply "transferred" to the next set of classes obtained with a higher $TH_{merge}$. When
increasing the merging threshold 0.45 → 0.50 the total number of classes rises 43 → 81 with 35 classes remaining identical
and 4 being "strongly similar" (Fig. 6);  only 4 classes from the original set remain without a strongly similar counterpart.
Further rising the threshold value 0.50 → 0.55 leads to building of 133 classes with 65 preserved identical and 5 "strongly
similar" to their counterpart (Fig. 6). The fulfilment of the demand on the consistency of class evolution is shown by the
prevalence of identical classes in the diagram, indicating one-to-one correspondence between sets of classes. The identical
classes, that remain unchanged, are accompanied by a 'bunch' of thin lines, which indicate that the left classes have some
similarity to the right classes. Such 'bunches' are mainly produced by splitting of classes on the left side into two or more (on
the right side). An unwanted form of the diagram would be a distribution of classes from set to set connected with thin lines,
without clearly preserved identical types.


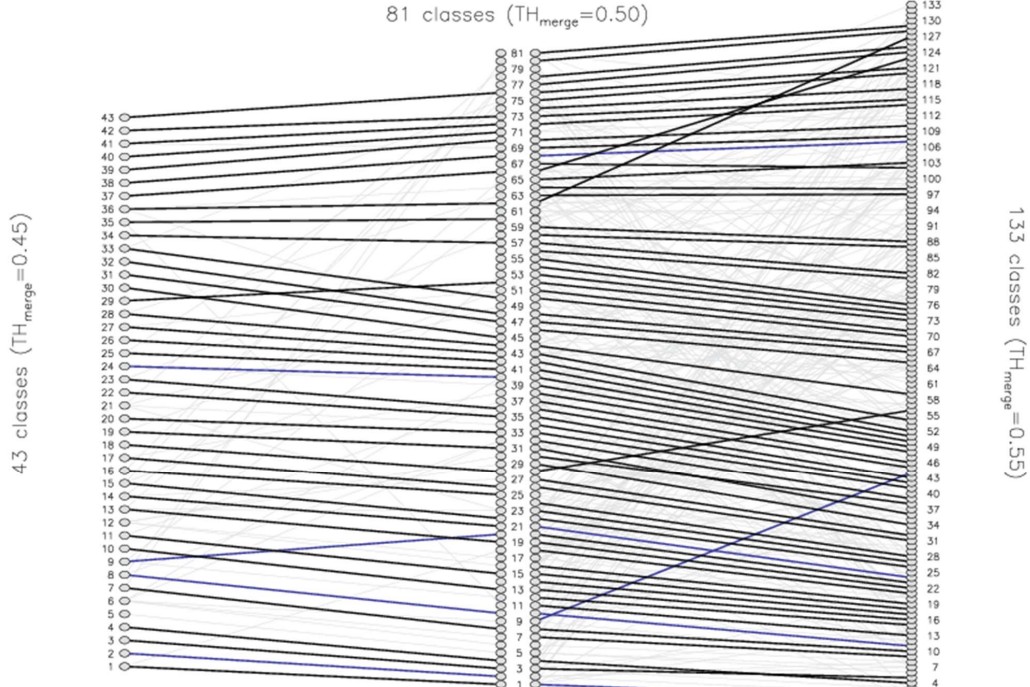

**Figure 6: Similarity between classes derived with different merging threshold: (left) 43 classes obtained with $TH_{merge}$=0.45, (middle) 81 classes with $TH_{merge}$=0.50, and (right) 133 classes with $TH_{merge}$=0.55. Numeration of classes is done for each class on the left; only every 2nd class in the middle and only every 3rd class on the right set of classes to avoid over-plotting. Black thick lines connect identical classes ($SSIM=1$), blue lines connect "strongly similar" classes ($0.75 \leq SSIM < 1$), grey lines connect classes with some similarity ($0.45 \leq SSIM < 0.75$).**

## 4.3 Cluster separability

From the setting of the stop-criterion in the clustering algorithm, the maximum similarity between medoids is less than $TH_{merge}$=0.45. In other words, there is no pair of final medoids similar to each other. Although it cannot be demanded that cluster centroids (means) also satisfy the same criterion on the maximum similarity pairwise, it can be demanded that cluster centroids are at least not "strongly similar" i.e. SSIM<0.75, for all pairs of centroids. Figure 7 shows the matrices of similarities for medoids and for centroids. Some pairs of centroids have similarity value higher than any pair of medoids (Fig. 7: circles show $SSIM>0.55$). This is due to the fact, that the similarity of medoids but not of centroids was the optimized criterion in the clustering algorithm. The maximal similarity for a pair of centroids is $SSIM=0.69$ (for centroids 14





and 18) i.e. there is no pair of "strongly similar" centroids. This gives an evidence that the two-stage clustering algorithm that uses medoids as class centres produces classes with also meaningfully separated centroids.

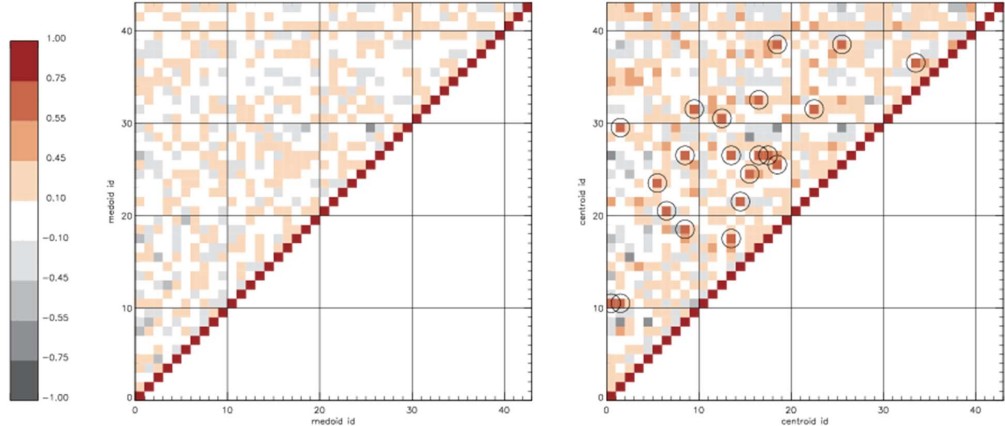


**Figure 7: Matrix of pairwise similarity values for 43 classes derived with $TH_{merge}$=0.45. Left panel: the matrix of SSIM-values for cluster medoids. Right panel: the matrix of SSIM-values for cluster centroids. Circles show similarity values greater than 0.55. Only upper left half of each matrix is shown because of the symmetry; diagonal elements have SSIM-value of 1.**

The metrics *EV*, *DRATIO*, and *FSIL* computed on the classes obtained with three different values of $TH_{merge}$ illustrate the

importance of the choice of this threshold and its influence on the number of derived classes and their separability. Table 1 presents the values of the chosen metrics. Please note: these metrics illustrate only (!) the influence of the $TH_{merge}$ on the final set of classes and do not describe the quality of classes as *EV*, *DRATIO*, and *FSIL* are computed using the Euclidean Distance – a measure that was not optimized by the clustering algorithm. Also *EV*, *DRATIO*, and *FSIL* should not be used to access the absolute performance of the classification, but the relative performance depending on the pre-set parameter

$TH_{merge}$.

**Table 1. Metrics for classes obtained in three experiments with varying merging-threshold ($TH_{merge}$) applied on the full 40 years of reference data. Values after "/" are those computed on random groups.**

| $TH_{merge}$ | Number of classes | *EV* classes/random | *DRATIO* classes/random | *FSIL* classes/random |
|---|---|---|---|---|
| 0.45 | 43 | 0.42/0.00 | 0.57/1.00 | 0.12/-0.13 |
| 0.50 | 81 | 0.47/0.01 | 0.53/0.99 | 0.15/-0.12 |
| 0.55 | 133 | 0.50/0.01 | 0.48/0.98 | 0.17/-0.12 |

Classifications with larger numbers of classes generally achieve a better skill (*EV*) than those with lower numbers due to the

natural fact that a larger number of classes captures a higher fraction of the variation: in the extreme case when the number



of classes k is equal to number of data n, the total variation is explained and $EV$=1. Therefore, it would be dangerous to favour classifications with larger numbers of classes based on this metric. $EV$ should only be used to measure the ratio to which a set of classes accounts for the dispersion in the given data set. In the present study, the set of classes obtained with $TH_{merge}$=0.55 provides the highest ratio of explained variation (0.50). Randomly chosen groups explain no variation (Table 1).

The metric $DRATIO$ can be interpreted as the mean distance of cluster elements within clusters $DI$ to the mean distance to elements of other clusters $DO$. In a randomly chosen set of groups the value of $DI$ and $DO$ are nearly equal and their ratio is close to 1.00 (Table 1). In the hypothetical case of the largest number of classes, when k = n, $DO$=0 and $DRATIO$=0. A value of $DRATIO$=0.50 means that the mean distance between cluster elements is half as large as the mean distance to elements of

other clusters. All three sets of classes provide a $DRATIO$ less than 0.60. Expectantly, the strictest threshold on the similarity for merging classes provides the tightest classes and, therefore, the lowest $DRATIO$.

The metric $FSIL$ can be interpreted as the mean normalized difference of distances between each data element to its own cluster $fa$ and to its nearest neighbor cluster $fb$. In the "best case" $fa$ is small, $fb>fa$ and $FSIL>0$. Note that $FSIL>0$ indicates the separation of the cluster from any randomly assigned clusters. In the "worst case" $fb<fa$ and $FSIL$ shows negative values.

All three sets of clusters provide positive values of this metric indicating the separation between each cluster and its nearest neighbor cluster.

### 4.4 Cluster stability

We run the two-stage clustering with $TH_{merge}$ of 0.45, 0.50 and 0.55 on the input data from one to 40 years and plot the number of classes against the amount of input data (Fig. 8, solid lines). Figure 8 illustrates the influence of the tightening the

requirement on similarity for building clusters: higher thresholds $TH_{merge}$ produce larger numbers of final classes. However, at the same time the higher $TH_{merge}$ also loosens the requirement to separation among classes (higher similarity between classes is possible).

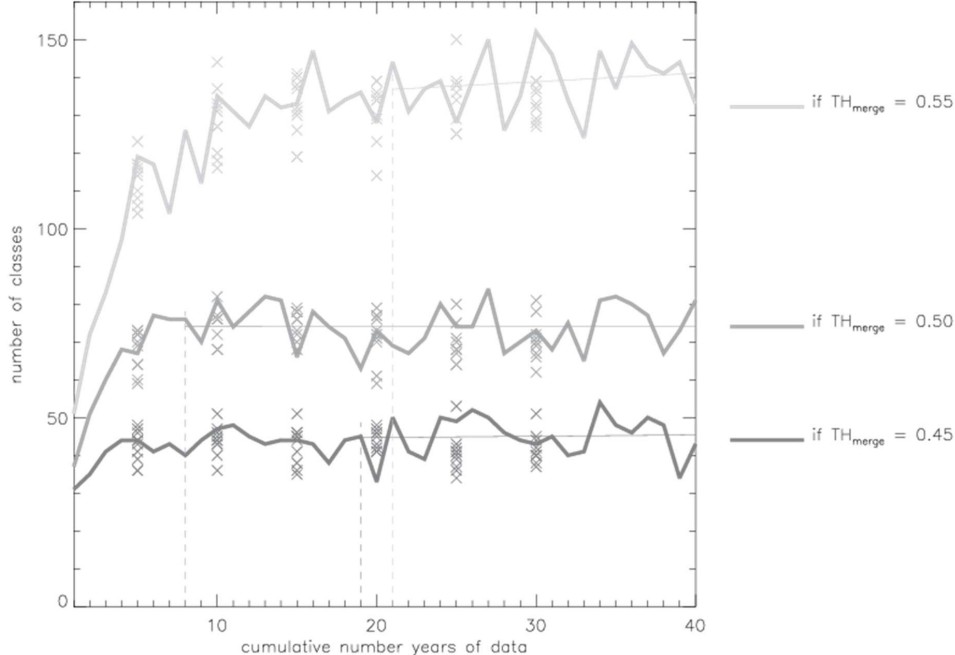

**Figure 8: Number of classes depends on the threshold *TH*<sub>merge</sub> and on the amount of clustered data. Horizontal lines show the**
**regression line with the min regression coefficient in a sliding window of 20 data points (each data point – one number of classes with corresponding data for 20 years). Crosses show number of classes build on alternative data volumes of 5, 10, 15, 20 and 25 years. Alternative periods were generated by sliding the time window by one year. The spread of the crosses illustrates the variation of number of classes for selected data volumes.**

The number of classes increases with the data volume in the beginning and then it levels out for all three tested values of the
threshold $TH_{merge}$ (Fig. 8). The routine to detect the minimal data volume after the number of classes stabilizes is the following. For the first 20 years of data the linear regression line is drawn through the corresponding 20 numbers of classes. The regression coefficient is saved for this step. For the next step, the regression line is drawn trough the next 20 numbers of classes (shifted by one year in the x-axis direction relative to the previous series). And so on until the last 20 numbers of classes for data volumes from 21 to 40 years are used. We obtain a series of 21 linear regression coefficients ordered by the
increasing data volume. We search for regression coefficients that have absolute value close to 0 i.e. in the interval [0.01, 0.01]; if there are several coefficients correspond to this criterion, we choose the one that relates to the smallest data amount. Finally, the related data amount (*x*-axis, vertical lines in Fig. 8) is the estimation of the min data volume for building the stable number of synoptic classes, the number that does not grow by adding more data into the clustering.



The threshold of 0.50 produces the stable number of classes already by 8 years of data. Other thresholds lead to a later

stabilization of the number of classes, after at least 19 and 21 years of data used. Therefore, we recommend using of at least

more than 21 years of data for building a representative set of SP-classes with the algorithm of this study.

The number of classes derived for more than 20 years of data is 45±5 (mean±standard deviation) with $TH_{merge}$=0.45, 73±6

and 138±8 for the higher values of merging threshold. The clustering method does not constrain the number of classes and,

therefore, shows some dependency on the input data as it may contain elements of different degree of similarity and result in

various numbers of final classes. However, this number of classes does not vary arbitrarily, but in a narrow range (Fig. 8)

that indicates its stabilization after more than 20 years of data is used.

For testing the stability of the method in space, additionally to the classes on the reference data set (2°x3°), further two sets of

classes were built: on the low-resolution (4°x6°) and on the high-resolution (1°x1.5°) data. Our clustering algorithm built 46

classes on the high-resolution data and 39 classes on the low-resolution data. The threshold on similarity for merging clusters

was the same as for the reference data $TH_{merge}$=0.45 (43 classes). This poses some restrictions on the interpretation of the

results. First: two images on different spatial resolutions derived from the same original image are not necessarily identical

(!) in terms of SSIM ($SSIM<1$) because they contain different amounts of information. The SSIM-value deteriorates with the

increasing spatial resolution as the degree of detail in the images grows. Following this argument, it would be impossible to

build the same set of classes at various spatial resolutions with the same threshold on similarity. However, it can be required

that some classes emerge at all spatial resolutions. Examples of such SP-classes are shown in Fig. 9 at all three spatial

resolutions.



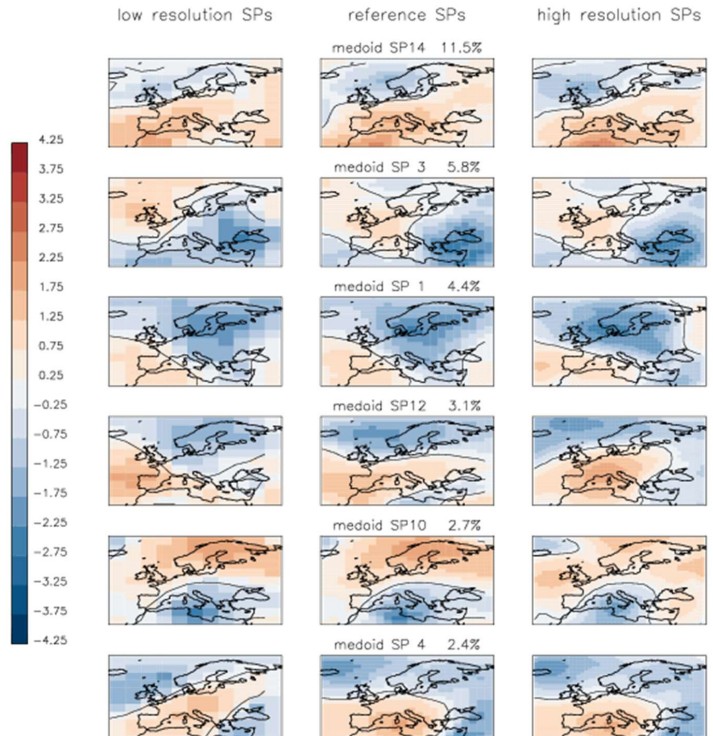

**Figure 9: Examples of SPs from the reference data (2°x3°, 22x22 grid cells) and corresponding low-resolution (4°x6°, 11x11 grid cells) and high-resolution (1°x1.5°, 44x44 grid cells) SPs obtained with the same threshold on cluster merging. The frequency of**
**each synoptic pattern in the reference data is given on the top of the reference SP plot.**

Figure 9 shows six SP-classes at the original resolution (centre plots) and their counterparts in the low- and high-resolution sets of classes. Please note: the SP-classes are built at each resolution independently and are not just re-sampled copies of the same classes. Therefore, some discrepancy must be tolerated among the classes at different resolutions as they are medoids of independently formed classes. Despite of such discrepancies the SP-classes show essentially the same synoptic situations
at all spatial resolutions.

### 4.5 Cluster reproduction and representativity

For each class the similarity value between its centroid and medoid is calculated. A good representativity is achieved when medoid and centroid of each class are "strongly similar" and $SSIM(medoid_i, centroid_i) \geq 0.75$ for all classes $i$. Figure 10 illustrates medoids and centroids for the five most frequent SP-classes. As expected, each medoid has a higher contrast of





anomalies and the corresponding centroid shows essentially the same pattern but with less contrast. The Mean Absolute
Difference (MD) between the two shows the highest values at the locations of strong amplitudes in the medoid fields and
lower values at locations on "edges" of synoptic patterns. This illustrates the strength of using the SSIM as a similarity
measure for pairs of geopotential fields: the clustering method sensitively groups SP-patterns with similar "edges" i.e. similar
composition of the anomalies.

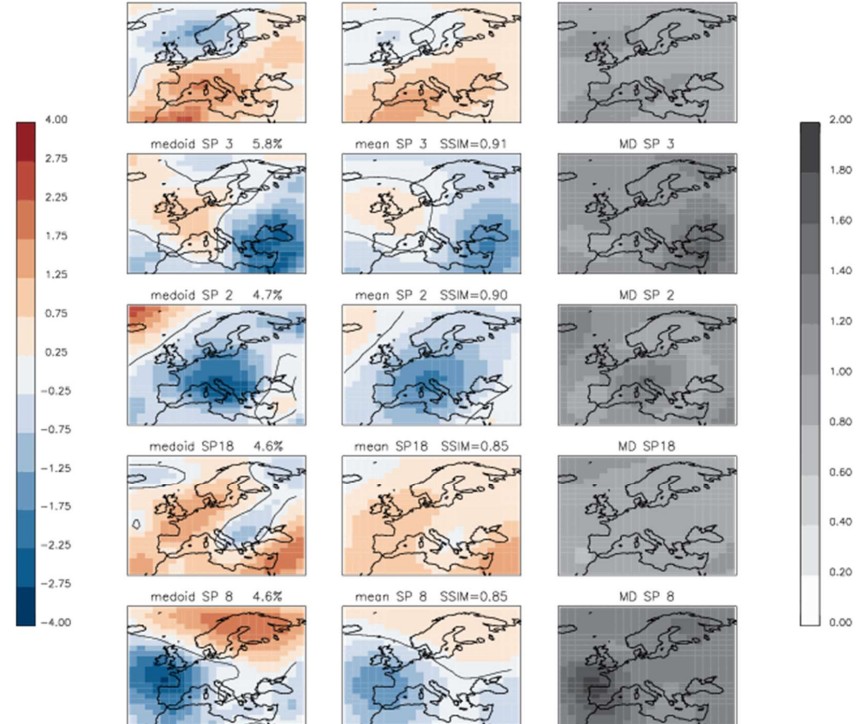


**Figure 10: Medoids (right), centroids (centre) and their Mean Absolute Difference (left) for five most frequent SP-classes. Index of
the SP-class is shown on top of each plot, SSIM between medoid and centroids is shown on the top of centroid plot.**

The similarity value between medoid and centroid for each class is computed and listed for all classes in the Table 2. The
"strong similarity" between medoids and centroids for all 43 classes was found indicating the very good representability of

clusters by their medoids. The mean similarity over all 43 classes is 0.84.



**Table 2: Classes of Synoptic Patterns (SP), number of elements (N) in the class, Fraction (Fr) in percent of the class in the reference data and the Similarity (SSIM) value between medoid and centroid of the class.**

| SP | N | Fr | SSIM | SP | N | Fr | SSIM | SP | N | Fr | SSIM |
|----|------|------|------|----|-----|-----|------|----|-----|-----|------|
| 1 | 649 | 4.4 | 0.89 | 16 | 239 | 1.6 | 0.80 | 31 | 71 | 0.5 | 0.82 |
| 2 | 688 | 4.7 | 0.90 | 17 | 374 | 2.6 | 0.89 | 32 | 124 | 0.8 | 0.77 |
| 3 | 849 | 5.8 | 0.91 | 18 | 679 | 4.6 | 0.85 | 33 | 310 | 2.1 | 0.82 |
| 4 | 347 | 2.4 | 0.86 | 19 | 315 | 2.2 | 0.82 | 34 | 124 | 0.8 | 0.81 |
| 5 | 532 | 3.6 | 0.84 | 20 | 216 | 1.5 | 0.87 | 35 | 194 | 1.3 | 0.87 |
| 6 | 404 | 2.8 | 0.86 | 21 | 209 | 1.4 | 0.82 | 36 | 210 | 1.4 | 0.76 |
| 7 | 314 | 2.1 | 0.84 | 22 | 141 | 1.0 | 0.82 | 37 | 99 | 0.7 | 0.86 |
| 8 | 674 | 4.6 | 0.85 | 23 | 109 | 0.7 | 0.82 | 38 | 301 | 2.1 | 0.83 |
| 9 | 404 | 2.8 | 0.82 | 24 | 151 | 1.0 | 0.81 | 39 | 171 | 1.2 | 0.81 |
| 10 | 389 | 2.7 | 0.88 | 25 | 253 | 1.7 | 0.87 | 40 | 198 | 1.4 | 0.89 |
| 11 | 208 | 1.4 | 0.84 | 26 | 274 | 1.9 | 0.87 | 41 | 54 | 0.4 | 0.85 |
| 12 | 446 | 3.1 | 0.85 | 27 | 426 | 2.9 | 0.86 | 42 | 94 | 0.6 | 0.79 |
| 13 | 434 | 3.0 | 0.89 | 28 | 199 | 1.4 | 0.85 | 43 | 105 | 0.7 | 0.82 |
| 14 | 1681 | 11.5 | 0.89 | 29 | 117 | 0.8 | 0.80 | | | | |
| 15 | 361 | 2.5 | 0.80 | 30 | 473 | 3.2 | 0.86 | | | | |

**4.6 Quality indices for CMIP6 historical climate simulations**

For each model, each statistic (*HIST*, *HIST$_{JFD}$*, *HIST$_{MAM}$*, *HIST$_{JJA}$*, *HIST$_{SON}$*, *TRANSIT*, *PERSIST*) is normalized to build the probability distribution (normalized to 1.0). Then the quality index *QI* between the two probability distributions is computed for each model and the reference. Table 3 shows individual and mean quality indices for all models. The mean quality index for a model is computed as the equally weighted mean of individual quality indices for the seven statistics of this model. A quality index of 1 indicates the identity of distributions; the quality indices for the alternative reanalysis are shown for comparison to the quality indices computed for models.



**Table 3: CMIP6 Models and their quality indices. For illustrating the quality of each model relative to the spread of multiple models, individual indices are highlighted in colour as follows: green shows "good" quality $QI_i > MEAN + STDDEV$, red shows**
**"bad" quality $QI_i < MEAN - STDDEV$, for each model $i$. MEAN and STDDEV are computed from the respective individual $QI$-values for all models. The mean quality index (Mean QI) is computed as the mean of individual quality indices for each model statistic.**

| Nr | Model name | QI(variable) | | | | | | | Mean QI |
| | | HIST | HIST$_{JFD}$ | HIST$_{MAM}$ | HIS$_{JJA}$ | HIST$_{SON}$ | TRANSIT | PERSIST | (all QIs) |
|---|---|---|---|---|---|---|---|---|---|
| - | ERAINT (reference) | 1.00 | 1.00 | 1.00 | 1.00 | 1.00 | 1.00 | 1.00 | 1.00 |
| - | NCEP1 (alternative) | 0.94 | 0.89 | 0.89 | 0.87 | 0.88 | 0.91 | 0.96 | 0.90 |
| 1 | ACCESS-CM2 | 0.93 | 0.70 | 0.87 | 0.80 | 0.83 | 0.87 | 0.93 | 0.85 |
| 2 | AWI-ESM-1-1-LR | 0.88 | 0.79 | 0.82 | 0.74 | 0.61 | 0.86 | 0.94 | 0.81 |
| 3 | BCC-CSM2-MR | 0.90 | 0.76 | 0.83 | 0.82 | 0.80 | 0.86 | 0.94 | 0.84 |
| 4 | BCC-ESM1 | 0.93 | 0.74 | 0.83 | 0.80 | 0.86 | 0.87 | 0.94 | 0.85 |
| 5 | CanESM5 | 0.90 | 0.72 | 0.84 | 0.78 | 0.78 | 0.85 | 0.94 | 0.83 |
| 6 | CESM2 | 0.91 | 0.78 | 0.88 | 0.82 | 0.66 | 0.87 | 0.94 | 0.84 |
| 7 | CESM2-FV2 | 0.91 | 0.73 | 0.87 | 0.81 | 0.67 | 0.87 | 0.94 | 0.83 |
| 8 | CESM2-WACCM-FV2 | 0.92 | 0.78 | 0.83 | 0.76 | 0.69 | 0.86 | 0.94 | 0.83 |
| 9 | CMCC-CM2-SR5 | 0.89 | 0.60 | 0.87 | 0.64 | 0.77 | 0.85 | 0.94 | 0.79 |
| 10 | CNRM-CM6-1 | 0.87 | 0.70 | 0.82 | 0.69 | 0.70 | 0.86 | 0.93 | 0.80 |
| 11 | CNRM-ESM2-1 | 0.88 | 0.73 | 0.82 | 0.68 | 0.70 | 0.86 | 0.92 | 0.80 |
| 12 | EC-Earth3 | 0.91 | 0.68 | 0.92 | 0.80 | 0.71 | 0.87 | 0.94 | 0.83 |
| 13 | EC-Earth3-Veg | 0.91 | 0.75 | 0.85 | 0.75 | 0.75 | 0.87 | 0.94 | 0.83 |
| 14 | FGOALS-f3-L | 0.87 | 0.63 | 0.75 | 0.72 | 0.85 | 0.85 | 0.93 | 0.80 |
| 15 | FGOALS-g3 | 0.89 | 0.58 | 0.79 | 0.78 | 0.85 | 0.85 | 0.92 | 0.81 |
| 16 | GISS-E2-1-G | 0.88 | 0.64 | 0.79 | 0.71 | 0.82 | 0.86 | 0.93 | 0.80 |
| 17 | HadGEM3-GC31-LL | 0.92 | 0.68 | 0.87 | 0.83 | 0.75 | 0.87 | 0.94 | 0.84 |
| 18 | HadGEM3-GC31-MM | 0.92 | 0.76 | 0.87 | 0.85 | 0.72 | 0.87 | 0.94 | 0.85 |
| 19 | INM-CM4-8 | 0.91 | 0.78 | 0.84 | 0.70 | 0.72 | 0.86 | 0.94 | 0.82 |
| 20 | INM-CM5-0 | 0.93 | 0.86 | 0.79 | 0.82 | 0.73 | 0.87 | 0.93 | 0.85 |
| 21 | IPSL-CM6A-LR | 0.88 | 0.80 | 0.81 | 0.68 | 0.65 | 0.85 | 0.93 | 0.80 |
| 22 | IPSL-CM6A-LR-INCA | 0.87 | 0.70 | 0.86 | 0.66 | 0.53 | 0.85 | 0.93 | 0.77 |
| 23 | KACE-1-0-G | 0.93 | 0.77 | 0.83 | 0.80 | 0.88 | 0.87 | 0.93 | 0.86 |
| 24 | MIROC6 | 0.88 | 0.83 | 0.89 | 0.52 | 0.82 | 0.86 | 0.93 | 0.82 |
| 25 | MPI-ESM-1-2-HAM | 0.92 | 0.78 | 0.83 | 0.82 | 0.76 | 0.87 | 0.94 | 0.85 |
| 26 | MPI-ESM1-2-HR | 0.94 | 0.82 | 0.87 | 0.85 | 0.84 | 0.88 | 0.95 | 0.88 |
| 27 | MPI-ESM1-2-LR | 0.90 | 0.72 | 0.86 | 0.79 | 0.74 | 0.86 | 0.93 | 0.83 |
| 28 | MRI-ESM2-0 | 0.94 | 0.76 | 0.85 | 0.76 | 0.88 | 0.87 | 0.94 | 0.86 |
| 29 | NorESM2-LM | 0.91 | 0.77 | 0.77 | 0.78 | 0.63 | 0.86 | 0.93 | 0.81 |
| 30 | NorESM2-MM | 0.93 | 0.82 | 0.87 | 0.81 | 0.71 | 0.88 | 0.94 | 0.85 |
| 31 | TaiESM1 | 0.93 | 0.73 | 0.91 | 0.74 | 0.78 | 0.88 | 0.94 | 0.84 |
| 32 | UKESM1-0-LL | 0.92 | 0.81 | 0.83 | 0.80 | 0.73 | 0.86 | 0.94 | 0.84 |
| - | *MEAN (all 32 models)* | 0.91 | 0.74 | 0.84 | 0.76 | 0.75 | 0.86 | 0.94 | - |
| - | *STDDEV (all 32 models)* | 0.02 | 0.07 | 0.04 | 0.07 | 0.08 | 0.01 | 0.01 | - |

The mean quality index indicates how well the respective model captures the synoptic circulation in the reference data ERA-
Interim. This quality index together with quality indices for scalar variables can be used for ranking the climate model simulations and as an evaluation measure. For example, the climate simulation MPI-ESM1-2-HR seems to outperform all

other models (Table 3) with the mean quality score of 0.88 that is close to the mean quality score of NCEP1-reanalysis (0.90). This model showed good individual quality indices for all individual model statistics (Table 3, all $QI$ are marked green except the quality index for histograms in spring months $QI(HIST_{MAM})$ indicating the "good" quality of the model).

There poorest mean $QI$=0.77 showed the model IPSL-CM6A-LR-INCA, that resulted from poor representation of the frequency of the synoptic classes (low $QI$ for individual model statistics $HIST$, $HIST_{JJA}$, $HIST_{SON}$) and from the poor representation of the transition frequencies ($TRANSIT$-matrix) i.e. incorrect representation of the sequence of the synoptic patterns. This diagnostic is a useful instrument to evaluate climate models, which gives an insight into the reasons for the poor model performance and the valuable feedback to model developers.

**5 Conclusions**

The presented two-stage clustering algorithm uses the Structural Similarity Index Measure (SSIM) for quantifying the similarity of synoptic circulation patterns pairwise. This measure mimics the perception of similarity by humans, is intuitively simple and computationally inexpensive. The use of SSIM in the clustering algorithm produces a set of well separated, consistent, stable and representative classes.

There is no "optimal classification" for all purposes and this set of classes is only one realization of multiple variants of other sets of classes that can be used to describe the atmospheric flow. The set of classes derived on some reference data, for example a reanalysis, may be used as the "reference set of classes for synoptic circulation". Statistical parameters (for example, the frequency of occurrence of each pattern or sequence of patterns) computed on some arbitrary climate simulation data attributed to these reference set of classes and compared to the parameters computed on the reference data

may be used for evaluation of climate simulations. Such parameters would quantify how well the synoptic circulation patterns are represented in the climate simulation data (as compared to the reference data) and provide additional diagnostic measures to the quality of these simulation.

**Acknowledgments**

Copyright "This service is based on data and products of the European Centre for Medium-Range Weather Forecasts

(ECMWF)".

**Competing interests**

The authors declare that they have no conflict of interest."



**Author Contributions**

Kristina Winderlich and Clementine Dalelane designed the methods. Kristina Winderlich developed the model code and
performed the simulations, prepared the manuscript with contributions from all co-authors.

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
