# Peer review of "Classification of synoptic circulation patterns with a two-stage clustering algorithm using the structural similarity index metric (SSIM)"

_Earth System Dynamics, 2022_

## Referee Comment (RC1)

Review of 'Classification of synoptic circulation patterns with a two-stage
clustering algorithm using the structural similarity index metric
(SSIM)'

In this paper the authors introduce a new clustering method for the analysis of synoptic weather
types over Western Europe, in a similar style to the traditional Grosswetterlagen approach. The
main novelties of the method are the use of the SSIM instead of Euclidean distance to compute
distances in the K-medoids algorithm, and a coupling of the K-medoids clustering to a hierarchical
agglomerative model which replaces the 'number of clusters' hyperparameter with a more intuitve
'maximum similarity' hyperparameter.
Using ERA-Interim reanalysis data they test the robustness of the method to parameter and
resolution variation, and show that it is essentially doing what they want it to do. Using these ERA-
Interim patterns, they then compute a number of metrics in CMIP6 models, and use this to make a
cursory assessment of model skill in representing synoptic European weather.
While I think the developed clustering method is interesting and has some potential benefits,
especially the clever 2-step procedure to find cluster number, I do not think the current manuscript
represents a strong research paper, and instead reads as more of a technical report. I have two main
issues:

1.  I do not think the analysis of CMIP6 simulations is very convincing, and in my opinion
    would need considerable extension to meet the stated aim of providing 'a useful instrument
    to evaluate climate models, which gives an insight into the reasons for the poor model
    performance and the valuable feedback to model developers.'
2.  Even if extended in this way, I do not believe the work fits well within the scope of ESD.
    To meet this scope, the work would in my view either need to engage with atmospheric
    dynamics (such as by investigating the drivers of good/bad synoptic pattern representation
    in CMIP models) or by exploring the socioeconomic impacts of their synoptic patterns
    (such as by looking at their relation to energy, agriculture, extreme event management,
    etc.). This would of course represent another major extension to the current work.

For these reasons, I unfortunately have to recommend the paper should be rejected as unsuitable for
ESD.
Below, I provide more detailed comments that may be of use to the authors in developing this work
further.

Detailed Comments

The choice to use a 22x22=484 dimensional space for cluster analysis is rather unusual, and bound
to add to the issues of instability, and low representativeness of the cluster means that you
comment on. Many approaches first reduce the phase space using EOF analysis, and are able to
capture >90 percent of the variability with <40 EOFs. It might be valuable to comment on why you
did not do this. Such approaches also reduce the 'structure insensitivity' of the standard Euclidean
distance metric by the way, as they preselect large scale modes that encode the spatial structure of
the flow.

The paper goes into considerable detail describing the new clustering method and demonstrating
various aspects of its robustness, with the reward being a new way of validating climate model
performance. However this most relevant aspect of the work is not explored in much detail, and
there are some issues with parts of the analysis that is present:
    •   The most important element of robustness has not been explored – robustness of the
        method to temporal variability. If we wish to use observationally identified patterns and

their statistics to evaluate the performance of uninitialised climate models, in either a historical or future context, then we must know how internal atmospheric variability alters the patterns and their statistics. While imperfect, there are many centennial reanalyses which could be used to look at synoptic patterns in different 40 year periods (as done in [1] for example). Failing this, a bootstrap approach could be used for the ERA Interim data. Without this, I find it very difficult to see how you can say that a low similarity between model and reanalysis SPs  is because the model is bad, rather than due to the SPs being properties of a very particular time frame.

- The TRANSIT and PERS metrics are based on the 42x42 transition matrix of the SPs which must surely be very noisy, with less than 8 datapoints on average for every element. In my experience looking at sets of <10 clusters, much more than 50 years of data are needed to even vaguely constrain transition matrix elements, especially ones representing rare transitions. I do not think these metrics can be telling you anything real about the skill of CMIP models.

[1] "Quantifying climate model representation of the wintertime Euro-Atlantic circulation using geopotential-jet regimes", Dorrington, Strommen and Fabiano 2022, Weather and Climate Dynamics, https://doi.org/10.5194/wcd-3-505-2022

---

## Author Comment (AC1)

We would like to ask the Editor to make the judgement, which options the manuscript may still have.

Below, we would like to address comments of Reviewer 1. For an easier reading, we highlight comments of Reviewer 1 in blue, our comment in black and citations from our paper in *black italic*.

We regret that Reviewer 1 suggests rejecting the manuscript. But we would like to draw editor's attention to the fact that this suggestion is based on a number of misunderstandings on the part of Reviewer 1 with regards to our manuscript:

1) "In this paper the authors introduce a new clustering method for the analysis of synoptic weather types over Western Europe, in a similar style to the traditional Grosswetterlagen approach" We did neither reproduce Grosswetterlagen nor tried to produce any set of other predefined synoptic circulations similar to them.

2) "The choice to use a 22x22=484 dimensional space for cluster analysis is rather unusual" We are aware of problems in the clustering of high-dimensional data, such as vague formulation of distance between data elements, correlation of some attributes of data element that may group them into different clusters. We solve this problem not by reducing the dimensionality of data, but by using the SSIM, that mimics human image-perception, instead of a classical distance measure (Lines 135-149) and by using the medoids (instead of centroids) for more stable representation of clusters (Lines 116-122).

3) ".. and bound to add to the issues of instability, and low representativeness of the cluster means that you comment on" We use medoids, not the means for building clusters for exactly the reasons of stability. Yes, we comment on low representativeness of cluster means (and therefor do not use them!) and discuss the choice of an alternative representation of clusters (Lines 116-122).

4) Reviewer 1 suggests as an option to extend the paper "… by exploring the socioeconomic impacts of their synoptic patterns (such as by looking at their relation to energy, agriculture, extreme event management, etc.)." Our manuscript presents a new clustering method, not its possible applications, for pre-selecting "good" models for subsequent applications such as impact-modelling etc.

5) Many approaches first reduce the phase space using EOF analysis, and are able to capture >90 percent of the variability with <40 EOFs. It might be valuable to comment on why you did not do this. We do discuss in detail exactly why we do not use traditional PCA-based techniques to initialize clusters (Lines 123-133).

6) In my experience looking at sets of <10 clusters, much more than 50 years of data are needed to even vaguely constrain transition matrix elements, especially ones representing rare transitions. I do not think these metrics can be telling you anything real about the skill of CMIP models. An objectively "optimal" number of classes for representing synoptic circulation does not exist [4]. The choice of this number strongly depend on the purpose of classification. About 1/3 of elements in the transition matrix contain the main load and contribute to the Quality Index most as we use the Jensen-Shannon distance (Equations 6-9, Pages 11-12). The choice of the Jensen-Shannon distance weights the contribution of each matrix element by its frequency (similar to computation of Kullback–Leibler divergence): frequent transitions govern contributions to the Quality Index, and vice versa, rare transitions make smaller contributions.

7) While I think the developed clustering method is interesting and has some potential benefits, especially the clever 2-step procedure to find cluster number, I do not think the current manuscript represents a strong research paper, and instead reads as more of a technical report. We introduce the new method for clustering synoptic patterns as an alternative to existing methods of clustering, which are performed on PCA-filtered data space. This novel approach allows accounting for rare synoptic situations, which may be linked to severe weather, and to avoid PCA-related deficiencies in pattern extraction discussed in detail in [1]. In our opinion, this alternative method bears its own scientific value, because as the very least it corroborates previous results, but it even improves upon those previous results in both statistical (number of classes is defined automatically) and climatological aspects (all data synoptic situations are classified). We demonstrate the application of the method for evaluating of CMIP6 models as an example.

**We appreciate the suggestion of Reviewer 1 to perform the analysis on robustness of the method to the temporal variability of the data. We are willing to include results of this analysis into the next version of the manuscript.**
* * *
Our answers to comments of Reviewer 1 in detail:

In this paper the authors introduce a new clustering method for the analysis of synoptic weather types over Western Europe, in a similar style to the traditional Grosswetterlagen approach.
This comment is misleading: we did neither reproduce Grosswetterlagen nor tried to produce any set of other predefined synoptic circulations. Our two-stage clustering method derived a set of synoptic circulation patterns automatically. Some of these synoptic patterns resemble already known Grosswetterlagen. This resemblance gives us an evidence that the method is able to find known synoptic patterns, not just some arbitrary circulations (Lines 314-323).

The main novelties of the method are the use of the SSIM instead of Euclidean distance to compute distances in the K-medoids algorithm, and a coupling of the K-medoids clustering to a hierarchical agglomerative model which replaces the 'number of clusters' hyperparameter with a more intuitive 'maximum similarity' hyperparameter.
We introduce a new method for classification of synoptic patterns without prior reduction of dimensionality (PCA-based, for example) and with a new similarity metric instead of classical distance-metrics.

Using ERA-Interim reanalysis data they test the robustness of the method to parameter and resolution variation, and show that it is essentially doing what they want it to do. Using these ERAInterim patterns, they then compute a number of metrics in CMIP6 models, and use this to make a cursory assessment of model skill in representing synoptic European weather.
We wonder why Reviewer 1 names the Quality Index, which we introduce, "a cursory assessment of model skill"? The Quality Index (Formulae 9, Line 305) itself was introduced in

[5] and can be computed on any similarity/distance measure. For Quality Index in this study we use the Jensen-Shannon distance measure computed on the frequency of synoptic patterns, their persistence and their transition matrix.

If this analysis is "cursory", we would appreciate if Reviewer 1 could make a suggestion on the analysis technique of models skill that is convincing.

While I think the developed clustering method is interesting and has some potential benefits, especially the clever 2-step procedure to find cluster number, I do not think the current manuscript represents a strong research paper, and instead reads as more of a technical report.

The paper presents the method for clustering synoptic patterns that differs from existing methods of clustering:

- It does not require reduction of data-dimensionality such as to PCA-filter because it uses new similarity metric SSIM
- It builds clusters stable to outliers as medoids (instead of centroids) represent classes
- It does not discard rare synoptic circulations, which may be important for further analysis i.e. extreme weather occurrence etc.
- It avoids PCA-related deficiencies in pattern extraction discussed in detail in [1]

We present one of possible applications for our classification (for illustrative purposes) – the evaluation of CMIP6 models as compared to the reference ERA-Interim over 1979-2015 period. We aim to provide the solid reference and documentation of this novel classification method, illustrating its application, for the broad scientific community.

I have two main issues:

1. I do not think the analysis of CMIP6 simulations is very convincing, and in my opinion would need considerable extension to meet the stated aim of providing 'a useful instrument to evaluate climate models, which gives an insight into the reasons for the poor model performance and the valuable feedback to model developers.'

The analysis of CMIP6 model simulations illustrates one of the possible applications of the method for the CORDEX-EU domain. The quality indices that we provide in the Manuscript (Table 3, Lines 473-478) show

1) how close the frequencies of synoptic patterns $QI(HIST)$ produces by CMIP6 models are to the Reanalysis
2) in which season of the year ($QI(HIST_{JFD})$, $QI(HIST_{MAM})$, $QI(HIST_{JJA})$, $QI(HIST_{SON})$) these frequencies are best reproduced
3) which CMIP6 model reproduces the persistence of synoptic patterns ($QI(PERSIST)$) best
4) how well the transition matrix is captured by CMIP6 models ($QI(TRANSIT)$)

We believe that findings as these, for example:

"a model X does not reproduce the correct frequency of SPs in summer"
"the transition matrix of SPs in model Y differs strongly from Reanalysis"
"a model Z fails to reproduce SP1 in winter" etc.

are valuable for model developers as they tell about particular deficiencies in the flow simulation by the models.

2. Even if extended in this way, I do not believe the work fits well within the scope of ESD. To meet this scope, the work would in my view either need to engage with atmospheric dynamics (such as by investigating the drivers of good/bad synoptic pattern representation in CMIP models) or by exploring the socioeconomic impacts of their synoptic patterns (such as by looking at their relation to energy, agriculture, extreme event management, etc.). This would of course represent another major extension to the current work.

We believe the manuscript indeed fits into the scope of the ESD journal because it contributes to the scope of the journal focused on investigations in the subject area 1."**Dynamics of the Earth system"** by a new concept for model evaluation in order to contribute to the model development and pre-selection for its further use such as future climate projections, impact-modelling and downscaling to smaller regions.

Reviewer 1 suggests an extension of the paper by either investigating "the drivers of good/bad synoptic pattern representation in CMIP models" or "exploring the socioeconomic impacts of their synoptic patterns". We consider both of these suggestions superfluous for the paper that presents an evaluation method for climate models. Firstly, because we believe that the aim of the evaluation routine is to find deficiencies in a model's performance and not to detect its reasons (for over 30 CMIP6-models it is also not feasible as these models differ in their grid, numerics, processes resolved and drivers used). Knowledge of model's deficiencies, which we quantify, would help model developers in their future work. Secondly, the evaluation of the performance of the climate model should ideally be done before impact-models are applied (and before the socioeconomic impacts are addressed with further impact-models). The main reason why we want to quantitatively access models performance independently on their subsequent application is to pre-select "the good ones".

For these reasons, I unfortunately have to recommend the paper should be rejected as unsuitable for ESD.
Below, I provide more detailed comments that may be of use to the authors in developing this work further.

Detailed Comments

The choice to use a 22x22=484 dimensional space for cluster analysis is rather unusual, and bound to add to the issues of instability, and low representativeness of the cluster means that you comment on.

The dimension space of the original data was reduced from the original ERA-Interim spatial resolution to the sampled 22x22 points (every 2° in latitude and every 3° in longitude directions over the CORDEX-EU domain, Figure 1 Page 4 of the manuscript) according to [3] for representing 500-hPa geopotential height at the synoptic-scale. We wonder why the size of 22x22=484 grid points is "bound to add to the issues on instability" as Reviewer 1 says. We show exactly the opposite in our paper. Increasing the spatial resolution of the classified fields to 44x44=1936 grid points and reducing it to 11x11=121 grid points yields essentially the same set of synoptic patterns (Figure 9, Page

20), which indicates the stability of the method to the horizontal resolution of the input data. *"Figure 9 shows six SP-classes at the original resolution (centre plots) and their counterparts in the low- and high-resolution sets of classes. Please note: the SP-classes are built at each resolution independently and are not just re-sampled copies of the same classes. Therefore, some discrepancy must be tolerated among the classes at different resolutions as they are medoids of independently formed classes. Despite of such discrepancies the SP-classes show essentially the same synoptic situations at all spatial resolutions"* (Lines 436-440).

Clustering of high-dimensional data poses two serious problems: 1) distance measure becomes less exact as the dimensionality grows and 2) data elements may share several correlated attributes that may group them in clusters differently. We solve both these problems by using the new similarity metric SSIM, that mimics human image-perception, instead of a classical distance measure (Lines 135-149) and by using the medoids (instead of centroids) for representing classes (Lines 116-122).

Yes, we comment on the low representativeness of cluster means and discuss the choice of an alternative representation of clusters in Lines 116-122 of the manuscript. In order to avoid the low representativeness of the cluster means we use medoids (not the means!) to represent clusters and show that the means and medoids of final classes are strongly similar (Figure 10, Page 21): *"The similarity value between medoid and centroid for each class is computed and listed for all classes in the Table 2. The "strong similarity" between medoids and centroids for all 43 classes was found indicating the very good representability of clusters by their medoids. The mean similarity over all 43 classes is 0.84"* (Lines 453-455).

Many approaches first reduce the phase space using EOF analysis, and are able to capture >90 percent of the variability with <40 EOFs. It might be valuable to comment on why you did not do this. Such approaches also reduce the 'structure insensitivity' of the standard Euclidean distance metric by the way, as they preselect large scale modes that encode the spatial structure of the flow.

We do discuss why PCA technique was not used for initialization of the clustering algorithm in the part "3 Method" of the manuscript *."* (In Lines 123-133*): "Decision 2: use a two-stage algorithm. There are multiple ways of defining the number of classes for a k-medoids algorithm (similarly to k-means) ranging from a random guess to the analysis of the data based on principal component analysis PCA, also known as empirical orthogonal functions, Huth (2000). Lee and Sheridan (2012) suggested the initialization of the clustering algorithm by selected PCAs. The reason for this statement was the common (naïve) assumption that the first few modes returned by PCA were physically interpretable and should match the underlying signal in the data. However, Fulton and Hegerl (2021) tested this signal-extraction method and demonstrated that it has serious deficiencies when extracting multiple additive synthetic modes: false dipoles instead of monopoles, which may lead to serious misinterpretation of extracted modes. Fulton and Hegerl (2021) also found that PCA tends to mix independent spatial regions into single modes. Therefore, we back off using the PCA-based initialization of the clustering algorithm and employ another classic clustering algorithm, hierarchical agglomerative clustering (HAC), for initializing the k-medoids."*

Additionally, using the PCA-based pre-filtering eliminates rare synoptic patterns from the analysis, but we want them to be included as 1) they represent variance of model dynamics, 2) their frequency of occurrence may change (rare synoptic patterns becoming more

frequent, for example, in the future) and 3) as they may be linked to extreme weather events and would be attributed to "common" synoptic patterns otherwise.

Reviewer 1 references to the study by Dorrington et al. (2022) on wintertime Euro-Atlantic circulations split in four main patterns, which investigates how well the tri-modal jet structure is represented by CMIP models. For this study, the small number of classes is important or even essential as it represents few a-priori known stable modes of circulation. In contrary, our clustering method does not have a purpose to identify few such modes. We do explicitly want to classify infrequent synoptic patterns in separate classes.

MSE has multiple serious disadvantages (it is insensitive to a contrast stretch, shift of means, contamination by Gaussian noise, etc.) as compared to structural similarity metrics when applied on data with temporal and spatial dependencies and on data where the error is sensitive to the original signal (discussed in detail and illustrated in [6]). Another alternative distance metric, Pearson correlation coefficient, is insensitive to differences in the mean and variance [2]. As we work with geopotential data that often reveal dependencies in time and space, as well as shifts in the mean and differing variances, we restrain from using "traditional" distance metrics and employ the structural similarity index measure (SSIM) widely used in digital video processing software.

The paper goes into considerable detail describing the new clustering method and demonstrating various aspects of its robustness, with the reward being a new way of validating climate model performance. However this most relevant aspect of the work is not explored in much detail, and there are some issues with parts of the analysis that is present:
• The most important element of robustness has not been explored – robustness of the method to temporal variability. If we wish to use observationally identified patterns and their statistics to evaluate the performance of uninitialised climate models, in either a historical or future context, then we must know how internal atmospheric variability alters the patterns and their statistics. While imperfect, there are many centennial reanalyses which could be used to look at synoptic patterns in different 40 year periods (as done in [1] for example). Failing this, a bootstrap approach could be used for the ERA Interim data. Without this, I find it very difficult to see how you can say that a low similarity between model and reanalysis SPs is because the model is bad, rather than due to the SPs being properties of a very particular time frame.

**We find this comment very valuable and are willing to include results of suggested analysis on robustness of the method on the temporal variability of the data into the manuscript.**

• The TRANSIT and PERS metrics are based on the 42x42 transition matrix of the SPs which must surely be very noisy, with less than 8 datapoints on average for every element. In my experience looking at sets of <10 clusters, much more than 50 years of data are needed to even vaguely constrain transition matrix elements, especially ones representing rare transitions. I do not think these metrics can be telling you anything real about the skill of CMIP models.

[1] "Quantifying climate model representation of the wintertime Euro-Atlantic circulation using geopotential-jet regimes", Dorrington, Strommen and Fabiano 2022, Weather and Climate Dynamics, https://doi.org/10.5194/wcd-3-505-2022

The frequent classes/transition-elements dominate the quality indices as we use the Jensen-Shannon distance (Equations 6-9, Pages 11-12): the influence of each signal to the final score is proportional to its frequency (similar to computation of Kullback–Leibler divergence). The usage of Jensen-Shannon distance makes the Quality Index most sensitive to the frequent classes/transition-elements and least sensitive to the "noise" from infrequent elements. There is no universally optimal number of classes to represent synoptic circulation in all applications. The choice of this number is often governed by the wish to reduce the numerical space of the subsequent analysis preserving the variance of the data in some degree. As the above mentioned study [4] tests "three standard numbers of types: A small one with 9, an intermediate one with 18 and a large number of 27. Even though these numbers might appear arbitrary, they represent the majority of the original classifications ..." Resulting number of classes in the present paper depends on the threshold of similarity between each pair of synoptic patterns: the higher the required similarity within each class, the larger number of classes will be built and vice versa.  If we would aim at building fewer classes (<10) as Reviewer 1 suggests, we would have either to loosen the requirement on the in-class similarity or to eliminate classes with fewer elements. However, we estimated similarity threshold experimentally based on the human perception and loosening this threshold would mean consciously grouping patterns, which are viewed by an observer as dissimilar, into one class. The elimination of infrequent synoptic classes we avoid on purpose: we do want to retain the rare classes as they may become more frequent in historical of future climate projections and may be linked to extreme weather.

References

[1] Fulton, D.J., Hegerl, G.C. (2021) Testing Methods of Pattern Extraction for Climate Data Using Synthetic Modes. Journal of Climate 34, 7645-7660.

[1] Mo, R., Ye, C., Whitfield, P.H. (2014) Application Potential of Four Nontraditional Similarity Metrics in Hydrometeorology. Journal of Hydrometeorology 15, 1862-1880.

[3] Muñoz, Á.G., Yang, X., Vecchi, G.A., Robertson, A.W., Cooke, W.F. (2017) A Weather-Type-Based Cross-Time-Scale Diagnostic Framework for Coupled Circulation Models. Journal of Climate 30, 8951-8972.

[4] Tveito, O.E., Huth, R., Philipp, A., Post, P., Pasqui, M., Esteban, P., Beck, C., Demuzere, M., Prudhomme, C., (2016) COST 580 Action 733 Harmonization and Application of Weather Type Classifications for European Regions.

[5] Sanderson, B.M., Knutti, R., Caldwell, P. (2015) A Representative Democracy to Reduce Interdependency in a Multimodel Ensemble. Journal of Climate 28, 5171-5194.

[6] Wang, Z., Bovik, A.C. (2009) Mean squared error: Love it or leave it? A new look at Signal Fidelity Measures. IEEE Signal Processing Magazine 26, 98-117

---

## Author Comment (AC2)

Comments of Reviewer 1 are in blue, answers of authors are in black/**black**.

Review of 'Classification of synoptic circulation patterns with a two-stage clustering algorithm using the structural similarity index metric (SSIM)'

In this paper the authors introduce a new clustering method for the analysis of synoptic weather types over Western Europe, in a similar style to the traditional Grosswetterlagen approach.

This comment is misleading: we did neither reproduce Grosswetterlagen nor tried to produce any set of other predefined synoptic circulations. Our two-stage clustering method derived a set of synoptic circulation patterns automatically. Some of these synoptic patterns resemble already known Grosswetterlagen. This resemblance gives us an evidence that the method is able to find known synoptic patterns, not just some arbitrary circulations (Lines 314-323).

The main novelties of the method are the use of the SSIM instead of Euclidean distance to compute distances in the K-medoids algorithm, and a coupling of the K-medoids clustering to a hierarchical agglomerative model which replaces the 'number of clusters' hyperparameter with a more intuitve 'maximum similarity' hyperparameter.

Yes, we introduce a new method for classification of synoptic patterns without prior reduction of dimensionality (PCA-based, for example) and with a new similarity metric instead of classical distance-metrics. The "similarity" parameter is intuitive as it is based on a human-perceived similarity of image pairs.

Using ERA-Interim reanalysis data they test the robustness of the method to parameter and resolution variation, and show that it is essentially doing what they want it to do. Using these ERAInterim patterns, they then compute a number of metrics in CMIP6 models, and use this to make a cursory assessment of model skill in representing synoptic European weather.

We wonder why Reviewer 1 names the Quality Index, which we introduce, "a cursory assessment of model skill"? The Quality Index (Formulae 9, Line 305) itself was introduced by Sanderson, et al. (2015) and can be computed on any similarity/distance measure. For Quality Index in this study we use the Jensen-Shannon distance measure computed on the frequency of synoptic patterns, their persistence and their transition matrix. Jensen-Shannon distance is computed on contributions of each "mismatch" between the model data and the reference weighted by its frequency (similar to Kullback–Leibler divergence) so that it is most sensitive to most frequent mismatches and least sensitive to rare ones.

If this analysis is "cursory", we would appreciate if Reviewer 1 could make a suggestion on the analysis technique of models skill that is convincing.

While I think the developed clustering method is interesting and has some potential benefits, especially the clever 2-step procedure to find cluster number, I do not think the current manuscript represents a strong research paper, and instead reads as more of a technical report.

We introduce the new method for clustering synoptic patterns as an alternative to existing methods of clustering, which are performed on PCA-filtered data space. We developed the method to suit our purposes: evaluation of climate models including rare synoptic situations. Our approach allows accounting for rare synoptic situations, which may be linked to severe

weather (who knows?!), and to avoid PCA-related deficiencies in pattern extraction discussed by Fulton and Hegerl (2021). Huth (2021) also demonstrated that still often used unrorated PCAs result in patterns that are rather artifacts of the analysis than true modes of variability. But the main reason why we restrain from using existing methods based on PCA-analysis is that they exclude rare synoptic situations deliberately taking only few PCAs with the largest load. This approach does not suit our purpose (as we want to account for rare synoptic situations too).

In our opinion, this our method is an alternative to existing methods and it bears its own scientific value, because as the very least it corroborates previous results, but it even improves upon those previous results in both statistical (number of classes is defined automatically) and climatological aspects (all data synoptic situations are classified). We demonstrate the application of the method for evaluating of CMIP6 models (for illustrative purposes) as compared to the reference ERA-Interim over 1979-2015 period. Our purpose is not to investigate each of the CMIP6 models individually for its performance but to provide a measure that ranks their relative performance (according to the chosen reference). The manuscript is written to document this, illustrating its application, for the broad scientific community.

I have two main issues:
1. I do not think the analysis of CMIP6 simulations is very convincing, and in my opinion would need considerable extension to meet the stated aim of providing 'a useful instrument to evaluate climate models, which gives an insight into the reasons for the poor model performance and the valuable feedback to model developers.'

The analysis of CMIP6 model simulations illustrates one of the possible applications of the method for the CORDEX-EU domain. The quality indices that we provide (Table 3, Lines 473-478) show
1) how close the frequencies of synoptic patterns $QI(HIST)$ produced by CMIP6 models are to the frequencies of these synoptic patterns in the Reanalysis
2) in which season of the year ($QI(HIST_{JFD})$, $QI(HIST_{MAM})$, $QI(HIST_{JJA})$, $QI(HIST_{SON})$) these frequencies are best reproduced
3) which CMIP6 model reproduces the persistence of synoptic patterns ($QI(PERSIST)$) best
4) how well the transition matrix is captured by CMIP6 models ($QI(TRANSIT)$)

We believe that findings as these, for example:
   "a model X does not reproduce the correct frequency of SPs in summer"
   "the transition matrix of SPs in model Y differs strongly from Reanalysis"
   "a model Z fails to reproduce SP1 in winter" etc.
are valuable as they tell about particular deficiencies in the flow simulation by the models and could be addressed by model developers.

2. Even if extended in this way, I do not believe the work fits well within the scope of ESD. To meet this scope, the work would in my view either need to engage with atmospheric dynamics (such as by investigating the drivers of good/bad synoptic pattern representation in CMIP models)

Our manuscript presents a new clustering method for pre-selecting "good" models for subsequent applications such as impact-modelling etc. We do not aim to present various evaluations of as many model as possible. The computation of Quality Indicies based on the chosen reference for synoptic classes for CMIP6 models is done for demonstrative purposes. Such computations can be done for any model and reference, depending on the evaluation

purpose of the user. We believe the manuscript indeed fits into the scope of the ESD journal because it contributes to the scope of the journal focused on investigations in the subject area 1."Dynamics of the Earth system" by a new concept for model evaluation in order to contribute to the model development and pre-selection for its further use such as future climate projections, impact-modelling and downscaling to smaller regions. We believe that the aim of the evaluation routine is to find deficiencies in a model's performance and not to detect its reasons (for over 30 CMIP6-models it is also not feasible as these models differ in their grid, numerics, processes resolved and drivers used). Knowledge of model's deficiencies, which we quantify, would help model developers in their future work.

> or by exploring the socioeconomic impacts of their synoptic patterns (such as by looking at their relation to energy, agriculture, extreme event management, etc.).

The evaluation of the performance of the climate model should ideally be done before impact-models are applied (and before the socioeconomic impacts are addressed with further impact-models). The main reason why we want to quantitatively access models performance independently on their subsequent application is to pre-select "the good ones".

This would of course represent another major extension to the current work.
We think that both of these suggestions are superfluous for a paper that presents a classification algorithm for synoptic situations with the aim of climate models evaluation.

For these reasons, I unfortunately have to recommend the paper should be rejected as unsuitable for ESD.

Below, I provide more detailed comments that may be of use to the authors in developing this work further.

Detailed Comments
The choice to use a 22x22=484 dimensional space for cluster analysis is rather unusual,
We are aware of problems in the clustering of high-dimensional data, such as:
1) distance measure becomes less exact as the dimensionality grows and
2) data elements may share several correlated attributes that may group them in clusters differently.
We solve both these problems by using the new similarity metric SSIM, that mimics human image-perception, instead of a classical distance measure (Lines 135-149) and by using the medoids (instead of centroids) for representing classes (Lines 116-122).

and bound to add to the issues of instability, and low representativeness of the cluster means that you comment on.
We use medoids, not the means for building clusters for exactly the reasons of stability. In a classical k-means clustering algorithms each cluster is represented by its mean. In our application such cluster mean is computed on multiple (often >600) synoptic maps (that are geopotential anomalies). This leads to a "smoothing" of such maps to a degree that the mean does not represent any realistic geopotential anomaly anymore, but a "blur" picture of some unidentifiable flow. The danger of using cluster means as cluster centers is that the "blur" centers attract multiple unsimilar elements into one cluster making it even more "blur". This effect is known as "snowballing". The final set of clusters is the rather small, but each cluster is likely to include elements strongly unsimilar to each other. This is the low

representativeness of cluster means we comment on in the manuscript (Lines 116-122). With the aim to avoid such "blur" cluster centers we discuss the choice of an alternative representation of clusters (Lines 116-122). In order to avoid the low representativeness of the cluster means we use medoids (not the means!) to represent clusters and show that the means and medoids of final classes are strongly similar (Figure 10, and Lines 453-455). This means: we used medoids (single elements for representing clusters) in the classification algorithm assuring that we avoid the "snowballing" and produced the final classes those cluster means resemble respective medoids. Therefore, using medoids is an efficient strategy for clustering and producing homogeneous clusters (clusters that only have elements that are similar to their centers).

Many approaches first reduce the phase space using EOF analysis, and are able to capture >90 percent of the variability with <40 EOFs. It might be valuable to comment on why you did not do this.

We do discuss in detail exactly why we do not use traditional PCA-based techniques to initialize clusters (Lines 123-133) in the part "3 Method" of the manuscript: *"Decision 2: use a two-stage algorithm. There are multiple ways of defining the number of classes for a k-medoids algorithm (similarly to k-means) ranging from a random guess to the analysis of the data based on principal component analysis PCA, also known as empirical orthogonal functions, Huth (2000). Lee and Sheridan (2012) suggested the initialization of the clustering algorithm by selected PCAs. The reason for this statement was the common (naïve) assumption that the first few modes returned by PCA were physically interpretable and should match the underlying signal in the data. However, Fulton and Hegerl (2021) tested this signal-extraction method and demonstrated that it has serious deficiencies when extracting multiple additive synthetic modes: false dipoles instead of monopoles, which may lead to serious misinterpretation of extracted modes. Fulton and Hegerl (2021) also found that PCA tends to mix independent spatial regions into single modes. Therefore, we back off using the PCA-based initialization of the clustering algorithm and employ another classic clustering algorithm, hierarchical agglomerative clustering (HAC), for initializing the k-medoids."*

The PCA-based pre-filtering technique does not suit our purpose because it eliminates rare synoptic patterns from the analysis. But we deliberately want the rare synoptic patterns to be included in the analysis for three reasons:
1) they represent variance of model dynamics,
2) their frequency of occurrence may change (rare synoptic patterns becoming more frequent, for example, in the future)
3) rare synoptic situations may be linked to extreme weather events that would be falsely attributed to frequent synoptic patterns otherwise.

Reviewer 1 references to the study by Dorrington et al. (2022) on wintertime Euro-Atlantic circulations split in four main patterns combined with diagnostics on how well the tri-modal jet structure is represented by CMIP models. For this study, the small number of classes is important or even essential as it represents few a-priori known circulation modes. In contrary, our clustering method does not aim to identify known modes, it does not aim to detect few of them. We do explicitly want to classify frequent and rare synoptic patterns in separate classes.

Such approaches also reduce the 'structure insensitivity' of the standard Euclidean distance metric by the way, as they preselect large scale modes that encode the spatial structure of the flow.

MSE has multiple serious disadvantages: it is insensitive to a contrast/amplitude stretch, shift of means, contamination by Gaussian noise, etc. as compared to structural similarity metrics when applied on data with temporal and spatial dependencies and on data where the error is sensitive to the original signal (as discussed in detail and illustrated by Wang and Bovik, 2016). Another alternative distance metric, Pearson correlation coefficient, is insensitive to differences in the mean and variance (Mo, et al., 2014). As we work with geopotential data that often reveal dependencies in time and space, as well as shifts in the mean and differing variances, we restrain from using above mentioned "traditional" distance metrics and employ the structural similarity index measure (SSIM) widely used in digital video processing software.

The paper goes into considerable detail describing the new clustering method and demonstrating various aspects of its robustness, with the reward being a new way of validating climate model performance. However this most relevant aspect of the work is not explored in much detail, and there are some issues with parts of the analysis that is present:
• The most important element of robustness has not been explored – robustness of the method to temporal variability. If we wish to use observationally identified patterns and their statistics to evaluate the performance of uninitialised climate models, in either a historical or future context, then we must know how internal atmospheric variability alters the patterns and their statistics. While imperfect, there are many centennial reanalyses which could be used to look at synoptic patterns in different 40 year periods (as done in [1] for example). Failing this, a bootstrap approach could be used for the ERA Interim data. Without this, I find it very difficult to see how you can say that a low similarity between model and reanalysis SPs is because the model is bad, rather than due to the SPs being properties of a very particular time frame.
**We find this comment very valuable and are willing to include results of suggested analysis on robustness of the method on the temporal variability of the data into the manuscript.**

• The TRANSIT and PERS metrics are based on the 42x42 transition matrix of the SPs which must surely be very noisy, with less than 8 datapoints on average for every element. In my experience looking at sets of <10 clusters, much more than 50 years of data are needed to even vaguely constrain transition matrix elements, especially ones representing rare transitions. I do not think these metrics can be telling you anything real about the skill of CMIP models.
[1] "Quantifying climate model representation of the wintertime Euro-Atlantic circulation using geopotential-jet regimes", Dorrington, Strommen and Fabiano 2022, Weather and Climate Dynamics, https://doi.org/10.5194/wcd-3-505-2022

Here we would like to remind that our study is not the first one used a seemingly large number of classes. A five-year (2005-2010) project named "Harmonisation and Applications of Weather Types Classifications for European Regions" (https://www.cost.eu/actions/733/) with participating research groups of 23 European countries produced an extensive catalogue of atmospheric circulation type classifications (cost733cat includes 17 automated classification methods and five subjective classifications, https://doi.org/10.1016/j.pce.2009.12.010) based on different methodological concepts,

algorithms and parameter options. This Action systematically evaluated an extensive number of classifications within a coordinated inter-disciplinary environment and presented the results in the final project repot by Tveito et al, (2016) downloadable here (https://opus.bibliothek.uni-augsburg.de/opus4/frontdoor/deliver/index/docId/3768/file/COST733_final_scientific_report_2016.pdf). One of the statements of this action is: there is no universally optimal number of classes to represent synoptic circulation in all applications. The choice of this number strongly depends on the purpose of the classification and is often governed by the wish to reduce the numerical space of the subsequent analysis preserving the variance of the data in some degree. As the above mentioned final project report (Tveito et al, 2016) tests "three standard numbers of types: A small one with 9, an intermediate one with 18 and a large number of 27. Even though these numbers might appear arbitrary, they represent the majority of the original classifications ...".  Participants of the COST733 showed a wide range of class numbers which, from few to over 40. The large number of classes is often used by methods rooted in synoptic meteorology, that give high priority to a high structural differentiation among synoptic patterns, at the same time trying  to maximize the homogeneity inside classes. This attempt results in some classes, which have a small number of members or could be even empty for a different time span.  On the other hand, methods that use a low number of classes may handle the pattern diversity in a sub-optimal way i.e. falsely attributing a pattern to a dissimilar class. None of these methods could be universally best suitable for all applications.

The number of classes in the present manuscript depends on the threshold of similarity between each pair of synoptic patterns: the higher the required similarity within each class, the larger number of classes will be built and vice versa.  If we would aim at building fewer classes (<10) as Reviewer 1 suggests, we would have either to loosen the requirement on the in-class similarity or to eliminate classes with fewer elements. However, we estimated similarity threshold experimentally based on the human perception and loosening this threshold would mean consciously grouping patterns, which are viewed by an observer as dissimilar, into one class. The elimination of infrequent synoptic classes we avoid on purpose: we do want to retain the rare classes as they may become more frequent in historical of future climate projections and may be linked to extreme weather. These are to important reasons for the "large" number of classes we aim to use for evaluating climate models and ranking them according to their performance relative to a given reference (reanalysis in our case).

Now back to the transition matrix. In our case, about ½  of all elements in the transition matrix contain the main load and contribute to the Quality Index most as we use the Jensen-Shannon distance (Equations 6-9, Pages 11-12). The choice of the Jensen-Shannon distance weights the contribution of each matrix element by its frequency (similar to computation of Kullback–Leibler divergence): frequent transitions govern contributions to the distance measure, and vice versa, rare transitions make smaller contributions (is least sensitive to the "noise" from infrequent elements).

References

Huth, R. & Beranová, R. (2021). How to recognize a true mode of atmospheric circulation variability. Earth and Space Science, 8, e2020EA001275.
https://doi.org/10.1029/2020EA001275

Fulton, D.J., Hegerl, G.C. (2021) Testing Methods of Pattern Extraction for Climate Data Using Synthetic Modes. Journal of Climate 34, 7645-7660.

Mo, R., Ye, C., Whitfield, P.H. (2014) Application Potential of Four Nontraditional Similarity Metrics in Hydrometeorology. Journal of Hydrometeorology 15, 1862-1880.

Muñoz, Á.G., Yang, X., Vecchi, G.A., Robertson, A.W., Cooke, W.F. (2017) A Weather-Type-Based Cross-Time-Scale Diagnostic Framework for Coupled Circulation Models. Journal of Climate 30, 8951-8972.

Tveito, O.E., Huth, R., Philipp, A., Post, P., Pasqui, M., Esteban, P., Beck, C., Demuzere, M., Prudhomme, C., (2016) COST 580 Action 733 Harmonization and Application of Weather Type Classifications for European Regions.

Sanderson, B.M., Knutti, R., Caldwell, P. (2015) A Representative Democracy to Reduce Interdependency in a Multimodel Ensemble. Journal of Climate 28, 5171-5194.

Wang, Z., Bovik, A.C. (2009) Mean squared error: Love it or leave it? A new look at Signal Fidelity Measures. IEEE Signal Processing Magazine 26, 98-117

---

## Author Comment (AC3)

Comments of Reviewer 2 are in blue, answers of authors are in black/**black**.

**Review of the manuscript entitled**: "Classification of synoptic circulation patterns with a two-stage clustering algorithm using the structural similarity index metric (SSIM)" by Kristina Winderlich, Clementine Dalelane and Andreas Walter

**Summary**
The authors develop a new classification method for synoptic circulation patterns with the aim to extend the evaluation routine for climate simulations. Its unique novelty is the use of the structural similarity index metric (SSIM) instead of traditional distance metrics for cluster building. This classification method combines two classical clustering algorithms used iteratively, hierarchical agglomerative clustering (HAC) and k-medoids. The authors apply the classification method to ERA-interim and NCEP1 reanalysis, and CMIP6 models. The authors wish to demonstrate that the built classes are consistent, well separated, spatially and temporally stable, and physically meaningful. Finally, the authors rank the CMIP6 models according to their ability to represent the weather types using different quality indices.

Dear authors,

The purpose of using synoptic circulation patterns to evaluate climate models is a welcomed aim, but is not the first time this is done, as it may seem from the text. Indeed, the ability of models to capture the characteristics of synoptic patterns is an important aspect of improving climate model simulations. The SSIM is generally an interesting and seems to be promising approach for the classification of weather regimes. The article is generally well written, however it should be extended to serve as a high quality research article in ESD.

We do not pretend to be the first in the field. Therefore, we often refer to the project named "Harmonisation and Applications of Weather Types Classifications for European Regions" (https://www.cost.eu/actions/733/) that finished by the time we started developing our method. This international project joined research groups of 23 European countries to produce an extensive catalogue of atmospheric circulation type classifications (cost733cat includes 17 automated classification methods and five subjective classifications, https://doi.org/10.1016/j.pce.2009.12.010) based on different methodological concepts, algorithms and parameter options. This Action systematically evaluated an extensive number of classifications within a coordinated inter-disciplinary environment and presented the results in the final project repot by Tveito et al, (2016) downloadable here (https://opus.bibliothek.uni-augsburg.de/opus4/frontdoor/deliver/index/docId/3768/file/COST733_final_scientific_repo rt_2016.pdf).

Pointing to this and other publications, we tried to shorten the introduction. It seems being cut too short. **We suggest extending the introduction of our manuscript by a new chapter, with an overview of existing synoptic classifications and their applications to model evaluation.**

My comments and suggestions to improve the manuscript are as follows:
**General comments**

- Many classification algorithms attempt to categorize weather types/regimes over the Atlantic-European-Mediterranean region. If the authors suggest a new procedure, they

should at least demonstrate why their classification is better than other classification procedures. Indeed, the authors try to explain their choices, but do not demonstrate how their procedure is superior in comparison to other classifications. Perhaps the authors can randomly select days and subjectively see for how many of them the classification does a decent job? Comparing to the original classification you mention in the text would then provide a semi-quantitative way of demonstrating the improvement from one classification to the other.

**We suggest including following parts into the manuscript:**

1) **an additional comparison of synoptic classes derived with a "standard" k-means routine for illustrating the advantage of using our k-medoids method.**
2) **application of our two-stage classification on different sets of randomly chosen data for comparing the resulting classes.**

- Forty-three classes seems a rather large number of weather types and can probably be significantly reduced by some sort of EOF analysis. If not, it should at least be explained why the authors do not use this approach as it is very common. Furthermore, I would like to see some further explanation on how do these synoptic types relate to the four canonical weather regimes.

Our choice of not using PCA-based pre-filtering of the data is explained in detail (Lines 123-133) in the part "3 Method" of the manuscript: *"Decision 2: use a two-stage algorithm. There are multiple ways of defining the number of classes for a k-medoids algorithm (similarly to k-means) ranging from a random guess to the analysis of the data based on principal component analysis PCA, also known as empirical orthogonal functions, Huth (2000). Lee and Sheridan (2012) suggested the initialization of the clustering algorithm by selected PCAs. The reason for this statement was the common (naïve) assumption that the first few modes returned by PCA were physically interpretable and should match the underlying signal in the data. However, Fulton and Hegerl (2021) tested this signal-extraction method and demonstrated that it has serious deficiencies when extracting multiple additive synthetic modes: false dipoles instead of monopoles, which may lead to serious misinterpretation of extracted modes. Fulton and Hegerl (2021) also found that PCA tends to mix independent spatial regions into single modes. Therefore, we back off using the PCA-based initialization of the clustering algorithm and employ another classic clustering algorithm, hierarchical agglomerative clustering (HAC), for initializing the k-medoids."*

Huth (2021) also demonstrated that (still often used!) unrorated PCAs result in patterns that are rather artifacts of the analysis than true modes of variability. Additionally, we suggest emphasizing in the text of the manuscript that the PCA-based pre-filtering technique does not suit our purpose because it eliminates rare synoptic patterns from the analysis taking only few PCAs with the largest load. This approach does not suit our purpose because we do want to include rare synoptic situations into separate classes.

We do deliberately want the rare synoptic patterns to be included in the analysis for three reasons:

1. they represent variance of model dynamics,
2. their frequency of occurrence may change (rare synoptic patterns becoming more frequent, for example, in the future)
3. rare synoptic situations may be linked to extreme weather events that would be falsely attributed to frequent synoptic patterns otherwise.

We introduce the new method for clustering synoptic patterns as an alternative to existing methods of clustering. Our classification method accounts for rare synoptic situations, which may be linked to severe weather (who knows?!), and avoids PCA-related deficiencies (for example, the extraction of bi-polar structures as an artifact by approximating a single-polar structure) in pattern extraction discussed by Fulton and Hegerl (2021).

In our opinion, this our method is an alternative to existing methods and it bears its own scientific value, because as the very least it corroborates previous results, but it even improves upon those previous results in both statistical (number of classes is defined automatically) and climatological aspects (all data synoptic situations are classified).

The reviewer uses the term "canonical" synoptic regimes probably inspired by inter-related studies of Fabiano et al. (2020) and Dorrington et al. (2022), who investigated the variability of the atmospheric circulation looking at four recurrent patterns: NAO+, NAO-, Scandinavian Blocking and Atlantic Ridge. This perspective of looking at the atmospheric circulation does not suit our aim of the model evaluation, because we do not focus on synoptic regimes (quasi-stationary states). The choice of the number of synoptic classes strongly depends on the purpose of the classification. In our manuscript, this number is rather large, as we do not eliminate infrequent classes as it is done by PCA-based classifications. The European COST Action 733 "Harmonisation and Applications of Weather Types Classifications for European Regions" (https://www.cost.eu/actions/733/) also says:  there is no universally optimal number of classes to represent synoptic circulation in all applications: "*three standard numbers of types: A small one with 9, an intermediate one with 18 and a large number of 27. Even though these numbers might appear arbitrary, they represent the majority of the original classifications ...*" (Tveito et al, 2016). Participants of the COST733 showed a wide range of class numbers from few to over 50.

- The CMIP6 model evaluation section in its current form is rather short and does not provide very useful information for model developers. This section should probably be extended. It would be nice to have some discussion as to why you think some models are better or worse. Additional analysis is of course welcomed, but should probably be balanced with the length of the article.

We use the CMIP6 models only for the illustrative purposes. The aim of our manuscript is to present a new classification method (as the title says; CMIP6 is not even mentioned in the title of the paper). We use CMIP6 models only to demonstrate how the models could be ranked according to the Quality Index computed on the reference set of synoptic patterns. In our opinion, such Quality Index should be used to extend the traditional set of metrics in model evaluation routine. A typical evaluation routine depends strongly on users demands and often includes bias-estimation and the analysis of extremes for scalar variables such as temperature and precipitation as stated in Lines 41-47 of OM: "*…traditional techniques for model evaluation mainly focus on individual variables and/or derived indices and do not take into account, how well models simulate synoptic weather patterns and their frequencies of occurrence (Díaz-Esteban et al., 2020)*".

Our Quality Index computed on the reference set of synoptic patterns gives additional information on model quality, firstly, to the users of the models supporting their choice for a model suitable for their applications and, secondly, to the model developers showing how well their model performs in comparison to other models. The bit of information that could be provided by our analysis to the model developers would sound like "Model X under-represents synoptic situation A in summer months" or "Model Y hast a strong mismatch in representing transition matrix in spring moths".

We believe that the aim of the evaluation routine is to quantify deficiencies in a model's performance and not to investigate/detect their reasons (for over 30 CMIP6-models it is also not feasible as these models differ in their grid, numerics, processes resolved and drivers used). This task can only be performed by the developers themselves. We would be happy to provide any modelling group with much more detailed evaluation results as to which synoptic pattern is malrepresented in which season to analyse possible deficiencies.

**Specific comments**
**Abstract**

- What do you mean with physically meaningful? There may be different meanings to physical, and you should probably clarify this in the text.

The "physically meaningful" synoptic class is a synoptic pattern known to exist in the data i.e. one that represents a realistic circulation.

- Line 10: This sentence should be at the very end of the abstract.

ok

- Do you think your classification would be useful for extended-range weather forecasts? If so, mention this and in the abstract and discuss in the conclusions.

As an optional application besides the model evaluation, a linkage of synoptic classes to extreme weather could potentially be addressed. For example, such linkage was used by Nguyen-Le and Yamada (2019), who classified anomalous weather patterns associated with heavy rainfall in Thailand and implemented classification results into a Global Spectral Model (GSM) of the Japan Meteorological Agency improving the forecast skill with the lead time up to 3-days. We are ready to extend the introduction/discussion part of the manuscript by such overview. However, we doubt that using synoptic classes in a form of "precursor" for improving a weather forecast beyond 3 days lead-time would be the best-suited instrument for improving such forecasts.

**Introduction**

- Line 43 – 47: From the introduction, it sounds as if you are the first and only group evaluating models based on weather regimes. However, there is an increasing body of knowledge working in this direction. To name a few articles:

References

Dorrington, J., Strommen, K., and Fabiano, F.: Quantifying climate model representation of the wintertime Euro-Atlantic circulation using geopotential-jet regimes, Weather Clim. Dynam., 3, 505–533, https://doi.org/10.5194/wcd-3-505-2022, 2022.
Fabiano, F., Christensen, H.M., Strommen, K. et al. Euro-Atlantic weather Regimes in the PRIMAVERA coupled climate simulations: impact of resolution and mean state biases on model performance. Clim Dyn 54, 5031–5048 (2020). https://doi.org/10.1007/s00382-020-05271-w
Hochman A, Alpert P, Harpaz T, Saaroni H, Messori G. 2019. A new dynamical systems perspective on atmospheric predictability: eastern Mediterranean weather regimes as a case study. Science Advances 5: eaau0936. https://doi.org/10.1126/sciadv.aau0936

We are certainly not the first to evaluate models accounting to weather regimes. Just to name a few: an exemplary study on CMIP5-CMIP6 model evaluation over multiple regional

domains by A. Cannon ([https://doi.org/10.1088/1748-9326/ab7e4f](https://doi.org/10.1088/1748-9326/ab7e4f)), a study by U. Riediger using weather types obtained with a threshold-based classification method for the Central Europe (DOI: 10.1127/0941-2948/2014/0519), a full set of more than 20 classification schemes inter-compared and described in the COST733 Action, and many others.

**We suggest extending the introduction with an overview of other applications of synoptic classification methods.**

- Line 58: Please discuss the number of regimes some more. There are a few articles focusing on this aspect in the literature. Some use two regimes (Wallace and Gutzler, 1981), others use four (Vautard 1990), six (Falkena et al., 2020) or seven (Grams et al., 2017) regimes. This is important as you use an outstanding number of 43.
References
Falkena, S. K., de Wiljes, J., Weisheimer, A., & Shepherd, T. G. (2020). Revisiting the identification of wintertime atmospheric circula-tion regimes in the Euro-Atlantic sector. Quarterly Journal of the Royal Meteorological Society, 146, 2801–2814. https://doi.org/10.1002/qj.3818
Grams, C. M., Beerli, R., Pfenninger, S., Staffell, I., & Wernli, H. (2017). Balancing Europe's wind-power output through spatial deployment informed by weather regimes. Nature Climate Change, 7, 557–562. https://doi.org/10.1038/nclimate3338
Vautard, R. (1990). Multiple weather regimes over the North Atlantic: Analysis of precursors and successors. Monthly Weather Review, 118,2056–2081. https://doi.org/10.1175/1520-0493(1990)118<2056:MWROTN>2.0.CO;2
Wallace, J. M., & Gutzler, D. S. (1981). Teleconnections in the geopotential height field during the Northern Hemisphere winter. MonthlyWeather Review, 109, 784–812. https://doi.org/10.1175/1520-0493(1981)109<0784:TITGHF>2.0.CO;2

There is no universally right number of synoptic classes for all applications. Each application requires a number of classes best suitable for its purposes. A large number of classes is often used by classification methods rooted in synoptic meteorology, that give high priority to a high structural differentiation among synoptic patterns, at the same time trying  to maximize the homogeneity inside classes. For example, the ZAMG-classification with 43 classes (Baur 1948, Lauscher 1985), and the Grosswetterlagen-based classification by James et al. (2007) with 58 weather types (29 for winter and 29 for summer). This attempt may produce some classes, which have a small number of members or could be even empty for a different time span.  On the other hand, methods that use a low number of classes may handle the pattern diversity in a sub-optimal way i.e. falsely attributing a pattern to a dissimilar class. None of these methods could be universally best suitable for all applications.

**We suggest adding a discussion part about the choice of number of classes into the manuscript.**

- Line 64-66: This is a very strong critic on all prior classifications and should be further explained why none fit your purpose. These classification procedures were all used extensively in the literature. If you state this, you should at least demonstrate how your classification is superior.

Our aim is to create an automated classification scheme that gives high priority to a structural differentiation among synoptic patterns, including rare patterns in separate classes (not leaving them unclassified and not attributing them into dissimilar classes).

**We suggest extending the part that explains why existing classifications do not suit for our purpose.**

**Data and methods**

- Line 80: If you use ERA-interim and not ERA5 reanalysis, you should at least say why, and mention some of the studies comparing the two data sets. I do not expect much difference for large-scale weather regimes, but this should be at least discussed.

At the time the work on the new classification started, the ERA-Interim was the data set with the largest temporal coverage (1979-2018) and commonly used as a reference data set for evaluating climate models. Certainly, the newer data set ERA5 could be used as the new reference, once available and routinely tested for "standard" evaluation applications such as bias-estimation and extreme statistics for models scalar variables.

- Line 82: Please justify why you use 12:00UTC and not daily or all 6-hourly data.
- Line 82: How did you coarse grain the data and why to 2×3 degrees?

Answer to both comments above: The spatial and temporal resolutions of the domain were chosen as recommended by the inter-comparison project COST Action 733 "Harmonization and Application of Weather Type Classifications for European Regions" (Tveito et al., 2016): every 2° in latitude and every 3° in longitude, daily at 12:00 UTC. The model output at 12:00 UTC is often chosen for evaluation for two practical reasons: 1) it often matches mid-day peak in extreme weather conditions and 2) is a typically available time for model output. The coarse-scale sampling of the geopotential field was also suggested by Muñoz et al., (2017) as sufficient due to the fact that the synoptic-scale 500-hPa geopotential height does not require high resolution to reproduce the key physical mechanisms associated with.

- You often use 'synoptic scale', but I think it is more accurate to consider these regimes as large-scale features. I would try being more accurate on this. Perhaps change throughout the text.

We use the term "synoptic scale" consciously addressing the weather-patterns that consist of positive and negative geopotential anomalies at a horizontal scale of about 1000 km seen together at a time. We treat all weather patterns equally independent on their temporal duration so that short-term patterns are deliberately included (not eliminated) into the classification. The term synoptic regime, from our point of view, describes rather a recurrent, quasi-stationary and temporally persistent state of the atmospheric circulation that can be associated, for example, with a NAO phase. **We do not aim to detect and classify such quasi-stationary regimes.**

- Line 95: Why 151 days of smoothing? Please justify this choice.

The 151-days smoothing with equal weighting to each element was done with the purpose to produce the very smooth seasonal curve of 500hPa-geopotential and its standard deviation. Such strong smoothing allows us to preserve as much as possible of the anomaly of the 500hPa-geopotential height after the normalization.

**Results**

- Lines 436-440: I do not completely understand how you obtained high resolution relative to coarse resolution in figure 9.

For testing the stability of the method in space, additionally to the classes on the reference data set (2°x3°), further two sets of classes were built by a resampling of the original data (the spatial resolution of the data set is approximately 80 km, T255 spectral): on the lowresolution (4°x6°) and on the high-resolution (1°x1.5°). This explanation should be added to the manuscript.

- Line 454-456: Your motivation was not to use centroids in the introduction and methods section, but then you test your medoids and say that they are very similar to the centroids. Is this not a circular argument?

No, it is not a circular argument. We use medoids, not the means for building clusters for the reasons of stability of the classification method. In a classical k-means clustering algorithms each cluster is represented by its mean. In our application such cluster mean is computed on multiple (often >600) synoptic maps (that are geopotential anomalies). This leads to a "smoothing" of such maps to a degree that the mean does not represent any realistic geopotential anomaly anymore, but a "blur" picture of some unidentifiable flow. The danger of using cluster means as cluster centers is that the "blur" centers attract multiple unsimilar elements into one cluster making it even more "blur". This effect is known as "snowballing". The final set of clusters obtained with such routine is rather small, but each cluster is likely to include elements strongly unsimilar to other elements (and we tested this indeed). This is the low representativeness of cluster means we comment on in the manuscript (Lines 116-122). With the aim to avoid such "blur" cluster centers we discuss the choice of an alternative representation of clusters (Lines 116-122). A medoid of a cluster can be seen as "the representative element" of this cluster i.e. element most similar to all other elements in the cluster. Once the cluster is changed (merged with another one by the hierarchical step for example) the medoids are recomputed. Every new attribution of an element is done to a cluster to whose medoid the element is the most similar. This ensures only attribution of similar elements to clusters and is called stability of the method.

Then, after the clusters are finally built using medoids, we show that also the cluster means are strongly similar (Figure 10, and Lines 453-455). This is not surprising as any new element attribution is done to the most similar medoid (similarity of a new cluster element to other elements is guaranteed in this case). This means: we used medoids (single elements for representing clusters) in the classification algorithm assuring that we avoid the "snowballing" and produced the final classes those cluster means resemble respective medoids. Therefore, using medoids is an efficient strategy for clustering and producing homogeneous clusters (clusters that only have elements that are similar to their centers).

- Section 4.6: Perhaps provide some illustrations of the different classes in the CMIP6 models, in addition the quality indices in the table.

The classification is only done on the reference reanalysis data. The only classes used in the evaluation are the classes (shown in Figure 4, Page 12) derived on these reanalysis data. CMIP6 model data are not used in the classification algorithm. The output of all CMIP6 model is assigned to these reference classes using the maximum similarity measure (SSIM).

Lines 284-289: "*The model output was assigned to the 43 reference classes derived from ERA-Interim and the following statistics were computed: histogram of frequencies (HIST) for SP-classes (year through), histograms of frequencies for each season (HIST$_{DJF}$, HIST$_{MAM}$, HIST$_{JJA}$, HIST$_{SON}$), matrix of transitions (TRANSIT) between available classes (frequency for each SP to follow another SP), and probability of persistence (PERSIST) of each SP for 1,2, .. 25 days. For each of these seven statistics an individual quality index (QI) is computed. The overall quality index is then computed as the mean of the seven individual quality indices.*"

If we would repeat the classification on each CMIP6 model output, it would produce different sets of synoptic classes not necessarily comparable.

- Table 3: I believe that there is not much difference between the models in the 'transit' and 'persist' values because there are so many classes. In addition, for the other indices the standard deviation is rather low, which is a bit surprising for more than 30 models. They all do pretty much the same job, which is again a bit surprising.

Absolute differences between models may be small. For the evaluation of these models their relative difference to the reference is important. As more classes are used the more differences between a model and a reference is captured (contributes to the Quality Index). Vice versa, with fewer classes – less differences are captured.

At best, in an extreme case, if we use just one single class – the difference would be characterized just by a single number.

- Are the models evaluation criteria significantly different from one another? I think you should test this.

The metrics to evaluate the classification method (chapter 3.3) are not independent; they are adapted form COST Action 733.

The Model evaluation criteria are independent. Although, the persistence and transition matrix may be viewed as dependent in some extent: the diagonal elements of the transitional matrix represent the probability of transition for each synoptic pattern to itself the next day, i.e. persistence, but not the duration of such transition. The persistence matrix represents the probability of duration of 1,2, 3 … days for each synoptic pattern.

**Conclusions**

- This section is rather very short and should have a bit more discussion with respect to other articles evaluating models using a classification procedure. The article would also benefit from explaining what is better or similar in the new classification with respect to other methodologies used in the literature. The potential use of this methodology in climate projections or extended-range weather forecasts should probably also be discussed.

**We are willing to add more discussion on the pointed topics.**

**Technical comments:**

- Line 82-84: Please rephrase, something is missing here.
- Line 307: This should be 'Results' and not 'Method' section.
- Line 318: Change 'gives us an evidence that' to 'provides evidence that'.
- Line 357: Change 'gives an evidence that' to 'provides evidence that'.

**Figures:**

- Figure 4: It is very hard to see anything with so many panels.
- Figure 10: I think you mixed up between left and right in the caption. In addition, are there significant difference in the right panels?
- Table 3: It should probably be DJF for winter in the upper row and not 'JDF'.

References

Baur, F., 1948: Einfüruhrung in die Grosswetterkunde (Introduction into Large Scale Weather), Dieterich Verlag, Wiesbaden, Germany

Hochman, A., Messori, G., Quinting, J. F., Pinto, J. G., and Grams, C. M.: Do Atlantic-European Weather Regimes Physically Exist?, Geophysical Research Letters, 48, 10.1029/2021gl095574, 2021.

Huth, R. & Beranová, R. (2021). How to recognize a true mode of atmospheric circulation variability. Earth and Space Science, 8, e2020EA001275. https://doi.org/10.1029/2020EA001275

James, P., 2007: An objective classi_cation method for Hess and Brezowsky Grosswetterlagen over Europe. Theor. Appl. Climatol. 88, 17/42.

Lauscher, F., 1985: Klimatologische Synoptik Osterreichs mittels der ostalpinen Wetterlagenklassifikation (Synoptic Climatology of Austria based on the Eastern-Alpine Weather Type Classi_cation). Technical report, Arbeiten aus der Zentralanstalt f ur Meteorologie und Geodynamik, Publikation Nr. 302, Heft 64, Austria

Muñoz, Á. G., Yang, X., Vecchi, G. A., Robertson, A. W., and Cooke, W. F.: A Weather-Type-Based Cross-Time-Scale Diagnostic Framework for Coupled Circulation Models, Journal of Climate, 30, 8951-8972, 10.1175/jcli-d-17-0115.1, 2017.

Tveito, O.E., Huth, R., Philipp, A., Post, P., Pasqui, M., Esteban, P., Beck, C., Demuzere, M., Prudhomme, C., (2016) COST 580 Action 733 Harmonization and Application of Weather Type Classifications for European Regions.

---

## Author Comment (AC4)

Comments of Reviewer 3 are in blue, answers of authors are in black/**black**.

This paper describes a novel method of clustering circulation fields, and then applies this method to assess the ability of CMIP6 models to simulate realistic circulation patterns.  The paper is generally clearly written and straightforward to understand, but I feel that the authors have not sufficiently justified the use of their method over something simpler like k-means.  The analysis of circulation in the CMIP6 models is also rather brief.  I therefore recommend major revisions.

Major comments

The bulk of the paper describes a new two-step classification method, arguing that previously used methods are 'suboptimal'.  However, I don't think that the authors have sufficiently motivated their choice of method - my suspicion is that standard k-means clustering would give similar results.

Using the medoids instead means for representing clusters was based on our simple experience: we constructed the two-stage clustering method using the k-means procedure (as the second step) and run it. In the application on 40 years of Reanalysis cluster means are computed on multiple (often >600) synoptic maps (that are geopotential anomalies). This lead to building of "smooth" maps to a degree that these means do not represent realistic geopotential anomalies anymore, but "blur" pictures of some unidentifiable circulations. We realized then that the danger of using cluster means as cluster centers is that the "blur" centers attract multiple unsimilar elements into one cluster making it even more "blur". This effect is often called "snowballing". The final set of clusters obtained with such routine is the rather small, but each cluster is likely to include elements strongly unsimilar to other elements (and we tested this indeed).

Therefore, with the aim to avoid such "blur" cluster centers, we discuss the choice of an alternative representation of clusters (Lines 116-122). A medoid of a cluster can be seen as "the representative element" of this cluster i.e. element most similar to other elements in the cluster. Once the cluster is changed (merged with another one by the hierarchical step for example) the medoids are recomputed. Every new attribution of an element is done to a cluster to whose medoid the element is the most similar. This ensures only attribution of similar elements to clusters.

**As answering to comments of Reviewer 2, we suggest including an additional comparison of synoptic classes derived with a "standard" k-means routine for illustration the advantage of using our k-medoids method.**

The authors argue that k-means clustering has a number of drawbacks:

i) the number of clusters has to be pre-specified.

(But the authors' similarity threshold parameter seems to play a similar role, as it is subjectively chosen and also influences the number of clusters.)

The threshold on similarity of a pair of images is based on human visual perception and can be estimated intuitively well. We estimated this threshold being th≈0.45 for similar pairs of geopotential anomaly images in the study domain asking 20 persons (our colleagues, 10 male and 10 female). As an example we show similar, weakly similar and dissimilar patterns in Figure 3, Page 9.

ii) k-means centroids could be misleading and unrepresentative of the fields in the cluster.

(But does this not also apply to medoids, as a single field chosen to represent a set of fields?  Surely any daily field will contain its own set of small scale features that don't resemble those of other fields.  The authors appear to find that the cluster centroids and medoids are pretty similar anyway.)

It is crucial to differentiate when medoids are used: in the classification algorithm (in the hierarchical clustering part and in the k-medoid part). Once the cluster is changed (merged with another one by the hierarchical step for example) the medoids are recomputed. Every new attribution of an element is done to a cluster with the most similar medoid. This ensures, at each iteration of the algorithm, only attribution of similar elements to clusters. After the clusters are finally built using medoids, we show that also the cluster means are strongly similar (Figure 10, and Lines 453-455). This means: we used medoids (single elements for representing clusters) in the classification algorithm assuring that we avoid the "snowballing" and produced the final classes those cluster means resemble respective medoids. Therefore, using medoids is an efficient strategy for clustering and producing homogeneous clusters (clusters that only have elements that are similar to their centers).

iii) k-means clusters could be sensitive to outliers.  (But does this actually happen in the case of the geopotential height fields?)

Yes, it does happen. Furthermore, it happens often. We did start using k-means in our algorithm and it produced fewer classes. But: multiple classes contained pairs of elements dissimilar to each other!  This is not surprising, as the classical k-means optimizes the within-class variance, that is the distance between the members and the mean, and does not (!) require elements of the class being similar to each other.

In a classical k-means algorithm, each cluster is represented by its mean. Such cluster mean computed on multiple (often >600) synoptic maps (that are geopotential anomalies) is often "smoothed" to a degree that it does not represent any realistic geopotential anomaly anymore, but a "blur" picture of some unidentifiable flow. The danger of using cluster means as cluster centers is that the "blur" centers attract multiple dissimilar elements into one cluster making it even more "blur". This effect is known as "snowballing". The final set of clusters obtained with such routine is rather small, but each cluster is likely to include elements dissimilar to other elements (and we have seen this indeed!). This is the low representativeness of cluster means we comment on in the manuscript (Lines 116-122). With the aim to avoid such "blur" cluster centers we discuss the choice of an alternative representation of clusters (Lines 116-122). A medoid of a cluster can be seen as "the representative element" of this cluster i.e. element most similar to other elements in the cluster. Once the cluster is changed (merged with another one by the hierarchical step for example) the medoids are recomputed. Every new attribution of an element is done to a cluster to whose medoid the element is the most similar. This ensures only attribution of similar elements to clusters and is called stability of the method.

**We suggest including an additional comparison of synoptic classes derived with a "standard" k-means routine for illustration the advantage of using our k-medoids method.**

The authors quote image processing references to justify the similarity metric used here over (say) mean square error.  It would be more convincing if the authors could show actual examples of deficiencies in k-means clusters constructed from their circulation

data, and/or that clusters produced using their method were superior to those produced using k-means (for example, using the criteria set out in section 3.3).

Choice of the SSIM metric. The choice of the SSIM instead of MSE was done at the very beginning of this work as the first classes were built that included pairs of elements visually dissimilar but with a rather small MSE. Therefore, we searched for a better measure to the structural similarity and turned to the field of image processing.

Example 1. The pairs of images (a,b) and (a,c) have nearly the same small MSE ( for comparison, the max MSE=11.6 for data of all daily data in 13 selected yeas) but are strongly different  in terms of SSIM:

*MSE(a,b)*=1.70 and *SSIM(a,b)*=0.71 → means images *a* and *b* are similar
*MSE(a,b)*=1.68 and *SSIM(a,b)*=-0.60 → means images *a* and *b* are dissimilar (SSIM <0!)

[Figure]

Example 2. The pairs of images (a,b) and (a,c):
*MSE(a,b)*=1.70 and *SSIM(a,b)*=0.68 → means images *a* and *b* are similar
*MSE(a,b)*=1.71 and *SSIM(a,b)*=-0.35 → means images *a* and *b* are dissimilar (SSIM <0!)

[Figure]

Choice of the cluster centers. **We suggest including an additional comparison of synoptic classes derived with a "standard" k-means routine for illustration the advantage of using our k-medoids method.**

2. The analysis of the CMIP6 models is rather limited - there's a ranking of the models

according to various metrics, but not much more.  Why did the authors choose these particular  metrics over the wide variety of other possibilities?

Evaluation methods range from single-variable to multi-variables biases, from climatic mean assessment to climate change (trends, periodicity, interseasonal/interannual/interdecadal variability), extreme values, abnormal values and quantitative evaluations of uncertainty for computed model variables. There are not many metrics in model evaluation besides those that use above-mentioned variables, such as, mean bias, extreme value statistics, and frequency of occurrence of a particular signal. We tried to choose a set of independent and informative metrics. In the present manuscript, we only introduce an additional Quality Index to be used in an evaluation routine (that may include additional metrics; this depends on requirements of the user that does the evaluation).

Do the HIST statistics correspond to biases in the mean state of the models?  Can the authors suggest any reasons why some models are better than others - eg resolution?

The histograms represent only the frequency of occurrence of synoptic patterns. From HIST we can see which patterns are under- and overrepresented. We expect models with shared/similar dynamical core doing similar errors.

However, we consciously restrain of judging possible reasons for better or worse model performance, as we believe that the aim of the evaluation routine is to quantify deficiencies. An investigation of reasons for a bad/good model performance would require good knowledge of all analyzed models (there over 30) regards  their grid, numerics, processes resolved and drivers used. Last, but not least, knowledge of a model's heritage (history of development) would be an important source for judging its performance as, for example, models sharing a similar dynamical core with the reanalysis model may have advantages as compared to models with different dynamical cores. These issues are not trivial to summarize and discuss in the scope of our humble manuscript about one clustering method.

Also, the transition statistics are likely to be very noisy with 43 different circulation types.   How can we be confident that the transition results from ERA-Interim are a meaningful benchmark - is there enough reanalysis data to do this?

"Noisiness" of statistics: The transitional matrix has 43x43 elements. In our case, about ½  of all these elements contain the main load (probability of transition > 0) and contribute to the Quality Index most as we use the Jensen-Shannon distance (Equations 6-9, Pages 11-12). The choice of the Jensen-Shannon distance weights the contribution of each matrix element by its frequency (similar to computation of Kullback–Leibler divergence): frequent transitions govern contributions to the distance measure, and vice versa, rare transitions make smaller contributions (it is quite robust against the "noise" from infrequent elements).

Benchmark: The Quality Index of NCEP1 (TRANS) is 0.91 – the higher value than any CMIP6 model gives  and additionally significantly overshooting the models with respect to the inter-model range of Quality Indices. This gives the confidence that the ERA-interim is meaningful benchmark.

Again, it would be interesting to know if the results of the model evaluation analysis are significantly different if k-means derived clusters are used instead.

We suggest demonstrating the clusters built by k-means. As discussed above, these classes contain dissimilar elements due to "snowballing" effects owing the use of the "blur" means for representing clusters. We chose representing of clusters by medoids in order to build clusters that are more homogeneous (contain elements similar to clusters medoid).

Line 49 - "Hochman et al proved" - I think 'proved' is only an appropriate word when discussing mathematical proofs.  I suggest something like 'argued' or 'demonstrated'.  Also, people arguing that clusters represent genuine low-frequency weather regimes tend to find relatively few of them (four in winter seems a popular choice).  Presumably the authors are not arguing that the 43 types they analyse here each represent a physical weather regime in this sense?

We do not aim to classify particularly low-frequency weather regimes. Our classification method was build with a purpose to include frequent as well as rare synoptic situations independently on their temporal occurrence and persistence. For searching low-frequency weather regimes other techniques such as PCA analysis (this approach "cuts off" PCAs with the largest load excluding other circulations as "noise") might be more suitable, not our approach.

Line 58 - 'the moving atmosphere' - I'm not sure what this means.
atmosphere in motion

Line 90 onwards - standardising the height fields means that information about the amplitude of the circulation anomalies is lost.  But different amplitude anomaly patterns could produce quite different responses in eg surface air temperature and precipitation, so I'm not sure the standardisation step is beneficial.

The normalization of the fields is necessary and essential as data for all seasons are classified. In the present manuscript, the geopotential fields are not linked to any weather phenomena such as temperature and/or precipitation.

line 111 - "The k-means clustering assigns every data element to the cluster center that is closest to it, if only by a small margin."  Isn't this true of any method that assigns each field to one of a set of a classes?

It is true for other methods, which do not employ probabilistic/fuzzy assignment to a class, as well. The focus of the sentence is on the phrase "if only by a small margin". K-means assigns all elements to the given number of classes. If this number is small, each element must be assigned to one class anyway regardless if there is any similarity to that class.  As the number of classes in k-means should be defined prior to the classification, the danger is high to produce classes with dissimilar (to each other) elements. Our method builds classes from only similar elements grouping them in the first step (merging in hierarchical step) and assigns each element to only similar class in the second step (k-medoids). The requirement of similarity of elements for building classes persists throughout the algorithm.

line 112 - "This makes the method sensitive to noise in the data and may lead to an assignment of a data element to a structurally dissimilar cluster center." - what does "structurally dissimilar" mean here?

A pair of images is structurally dissimilar when it shows patterns perceived by an observer (or characterized with any structural similarity measure) as dissimilar.

How can we distinguish the noise from the structure in any given field?
Using the SSIM index as it is constructed from three parts (structure/covariance, luminance/mean, contrast/variance) in order to detect the similarity between two images based on their shapes (covariance), match in intensity (mean) and structure/noise (variance).

The noise/variance of an image, which can be understood as "roughness" or "texture" of the image. is included into a part of SSIM.

Can the authors show examples of fields that are far apart under the Euclidean distance metric but close together under the similarity metric, or vice versa?

An example of pairs of images *(a,b)* and *(a,c)* with different MSE to the reference *a* and similar SSIM: both images *b* and *c* have the same structural similarity to the reference *a*, but the MSE is different.

[Figure]

An example of pairs of images with similar MSE and different SSIM: both images *b* and *c* have similar MSE to the reference image *a*. But image *b* is structurally similar to *a*, *c* – dissimilar.

[Figure]

line 116 - Doesn't using medoids also risk inflating the significance of small-scale noise in the daily field chosen as the medoid?
No, it does not as the "small scale noise" is only a part of the structural similarity measure (variance as contrast).

line 137 - "Wang and Bovik (2009) demonstrated that the MSE has serious disadvantages when applied on data with temporal and spatial dependencies" - dependencies on what?  Does this mean temporal and spatial correlations?

MSE remains the standard criterion for the assessment of signal quality and fidelity. It is widely used in optimization routines. However, MSE is not always a good measure to evaluate signals fidelity because it is insensitive such distortions of the original signal/image as: "*a contrast stretch, mean luminance shift, contamination by additive white Gaussian noise, impulsive noise distortion, JPEG compression, blur, spatial scaling, spatial shift, and rotation*" (See Figure 2 and Figure 7 in Wang and Bovik (2009), DOI: 10.1109/MSP.2008.930649). Using MSE is justified when evaluated patterns are 1) independent temporally and spatially, 2) the error of the signal is independent of the mean signal, 3) the sign of the error plays no role in the evaluation of the signal, 4) all errors are equally important in the evaluation of the original signal. None of these assumptions holds when we process geopotential height data, temperature, precipitation, surface pressure etc.

line 194 - is the similarity between two clusters measured using their medoid fields?
Yes.

line 267 - Is the algorithm stable if applied to slightly different initial subsets of the data? The number of patterns may be stable, but do the same patterns emerge from the clustering?
Similar patterns, not exactly the same, with similar frequencies.
**We suggest including a part into the manuscript that shows derived classes on randomly chosen sets of data.**

Figure 3 - it would make more sense to have the transition between the blues and reds in the colour bar at zero, not +0.25.
Yes, we suggest adding more contour lines.

Line 245 - should there be a reference to figure 6 here?
Yes, it was "lost"

Line 282 - "However, it is necessary to demand that a cluster medoid represents all cluster elements and their whole entity as a group." Does comparing the mediod and centroid really guarantee this?
Representing a cluster by a medoid guarantees that the medoid has a minimum similarity to each of the cluster elements, furthermore, it is the element with the largest total similarity to all of cluster elements. In other words, medoid is the representative element of the class. If the medoid and the centroid are similar, it guarantees that there are no or negligibly few "extravagantly" dissimilar members of that class. Otherwise, the mean (centroid) would have lost its similarity to the medoid distorted by the averaging of dissimilar members.

Line 307 - is section 4 meant to be labelled 'Method', the same as section 3?
4. Results.

Figure 4 - Can the colour bar be included in the figure? There's room in the bottom row of panels.
*The caption of the Figure 4 says "The legend for colour shading is the same as in Fig. 3.*" It can be repeated, yes.

Line 320 - "This correspondence gives us an evidence that, albeit not tuned to and not required to mimic semi-manual classifications, the new classification method determines

not just arbitrary synoptic patterns but those described by experts in semi-manual classifications."

I'm not convinced - given that there are 43 different types, it seems quite likely that some of them could resemble Grosswetterlagen patterns by chance.
Of course the only seemingly alike looking images of a derived synoptic class and a Grosswetterlage could be arbitrary. But we compared our classes to those of James et al. (2007), which have also a similar frequency of occurrence i.e. we detect similar patterns with similar frequency of occurrence. These frequencies can be added to the text.

Figure 7 - the text in the figure labels could be much larger for legibility.
We agree

line 447 - again, I don't think one can infer that this is an inherent advantage of the SSIM method without making a comparison with other cluster methods.
**We suggest including an additional comparison of synoptic classes derived with a "standard" k-means routine, that uses MSE distance, for illustration the advantage of using our k-medoids with SSIM method.**

---

## Author Comment (AC5)

Dear Editor Gabriele Messori,

Thank you for coordinating the second round of review for our manuscript.

Although we appreciate comments of the Reviewer 1, we are very disappointed about comments of Reviewer 3.

Reviewer 1: *"To summarise, I suggest that the authors more tightly focus the structure of the article around the importance of handling rare synoptic conditions and extremes in clustering approaches, showing an example situation where an impactful event was linked to a very rarely occurring circulation as motivation. I would then suggest a concrete demonstration that the EOF Kmeans with MSE approach more poorly handles rare circulations than the SSIM approach in ERA Interim. I think this, followed by the various robustness tests and the first look at CMIP6 already present would then make any reader keen to see this approach explored further. I would also remove or substantially reword some of the criticisms of PCA based clustering that are unrelated to extreme circulations for the reasons I discuss above."*

Reviewer 1 suggests to draw more attention to rare synoptic conditions and extreme weather. We agree this would improve the paper and stronger justify purposes of constructing the new classification algorithm. We also agree to shorten and rewrite our argumentation about existing PCA-based methods.

Unfortunately, comments of Reviewer 3 indicate that we were not convincing with our arguments in his/her opinion. He/she doubts about usefulness of our tests of classification methods on the synthetic data (*"I'm not clear why the authors are using synthetic Gaussian data to compare k-means and their k-medoid method"*), despite our explanation in Lines 152-157 in the manuscript: "*The first dataset, a dataset of synthetic data, is used to demonstrate the performance of the classification method explaining why modifications to the classical k-means algorithm are necessary. We generated this synthetic data set using Gaussian shaped anomalies … to illustrate how such anomalies are treated by the classification algorithm.*" The anomalies in synthetic data initially have circular shapes. We demonstrate how k-means destroys such shapes producing "distorted" (due to averaging) class centers. This effect is important to keep in mind when classifications are applied on geopotential fields: classes retrieved with k-means may show unrealistic geopotential and be non-interpretable. Our k-medoid based classification overcomes this shortcoming.

Furthermore, Reviewer 3 seems to doubt our honesty as he/she says *"I find this unconvincing - presumably the most dissimilar members are shown in the figure…"* as we show dissimilar class members in Figure 5 as a result of using MSE as similarity metric. The shown fields are just the first 15 members of the class, not deliberately chosen or pre-processed in any way for showing the failure of MSE-metric more critically as it is. Figures 3 and 5 show that MSE metric does not suit as similarity measure for our data: fields dissimilar to each other are grouped into one class. The reason for this is in the formulation of MSE – it does not account for correlation of patterns that plays an important role for grouping highly structural data. In contrast to MSE, our classification that uses SSIM does.

We wonder about the following comment from Reviewer 3 *"Finally, there is no corresponding comparison of dissimilar fields in one of the larger k-medoids clusters shown in panel 4d - do these clusters suffer from similar issues?"* as there are no dissimilar fields in classifications that use SSIM because weakly correlated fields yield a negative/very low values of SSIM and are not grouped into the same class. This follows from the definition of SSIM that includes the covariance term!

Reviewer 3: *"l295 I don't follow why having a cluster with weak anomalies would then attract more fields than other clusters with stronger anomalies"*.

This so-called snowballing effect results from the averaging of multiple class elements (= k-means classification) – see explanations in Lines 286-300 of the manuscript. This leads to iteratively weakening structures in class centers i.e. the more elements are assigned to this class the more dissimilarity between each class element to the class center is tolerated by the algorithm.

We also would like to add here: we do not re-discover deficiencies of k-means clustering with MSE as distance metric. Those deficiencies are widely known and already referenced in our manuscript. See for example "Finding Groups in Data. An Introduction to Cluster Analysis by L. Kaufman and P.J. Rousseeuw or other handbooks. More examples and discussion on the problem of use MSE as similarity metric can be learned from an outstanding paper of Wang and Bovik "Mean squared error: Love it or leave it? A new look at Signal Fidelity Measures" in IEEE Signal Processing Magazine, 26, 98-117, 10.1109/msp.2008.930649, 2009.

Regrettably we see no further way to convince Reviewer 3 with our argumentation and would like to withdraw our manuscript with an option (Option B) to resubmit a rewritten manuscript for an independent review and discussion, and possible publication in ESD at later time.

Sincerely,

Kristina Winderlich and Co-authors